# Integrins regulate hERG1 dynamics by girdin-dependent Gαi3: signaling and modeling in cancer cells

Claudia Duranti[1],*, Jessica Iorio[1],*, Giacomo Bagni[1], Ginevra Chioccioli Altadonna[1,7] , Thibault Fillion[2,3], Matteo Lulli[4], Franco Nicolas D'Alessandro[1,7], Alberto Montalbano[1] , Elena Lastraioli[1,8] , Duccio Fanelli[2,8], Stefano Coppola[5], Thomas Schmidt[5], Francesco Piazza[2,3,8], Andrea Becchetti[6],† , Annarosa Arcangeli[1,8],† 

The hERG1 potassium channel is aberrantly over expressed in tumors and regulates the cancer cell response to integrin-dependent adhesion. We unravel a novel signaling pathway by which integrin engagement by the ECM protein fibronectin promotes hERG1 translocation to the plasma membrane and its association with β1 integrins, by activating girdin-dependent Gαi3 proteins and protein kinase B (Akt). By sequestering hERG1, β1 integrins make it avoid Rab5-mediated endocytosis, where unbound channels are degraded. The cycle of hERG1 expression determines the resting potential ($V_{rest}$) oscillations and drives the cortical f-actin dynamics and thus cell motility. To interpret the slow biphasic kinetics of hERG1/β1 integrin interplay, we developed a mathematical model based on a generic balanced inactivation–like module. Integrin-mediated cell adhesion triggers two contrary responses: a rapid stimulation of hERG1/β1 complex formation, followed by a slow inhibition which restores the initial condition. The protracted hERG1/β1 integrin cycle determines the slow time course and cyclic behavior of cell migration in cancer cells.

## Introduction

A cell's response to the extracellular environment is regulated by a complex interplay of diffuse and local signals, which are often determined by macromolecular complexes that form on the plasma membrane in response to microenvironmental cues (Becchetti et al, 2022). Integrin adhesion receptors are central constituents of such complexes, as they link the cell to specific proteins of the ECM (Doyle et al, 2022). Cells integrate the ECM signals with those

conveyed by diffusible factors, through the formation of supramolecular complexes that give rise to integrin-centered signaling hubs (Giancotti & Tarone, 2003; Cabodi et al, 2010a; Humphrey et al, 2014). As is well known, integrin receptors transmit their signals bidirectionally (Hynes, 2002). In the "inside-to-out" mode, intracellular signals modulate the integrin's adhesive function, thanks to molecular complexes comprising talin, kindlin, and paxillin, as well as the actin cytoskeleton (Ross et al, 2013). In the "outside-to-in" mode, the classical signaling mechanism envisages the tails of integrin β subunits to associate with cytoskeletal, adaptor, and signaling molecules such as FAK and phosphatidylinositol-3-kinase (PI3K) (Campbell & Humphries, 2011). This is followed by the activation of tyrosine kinase receptors, P-type G proteins (Cabodi et al, 2010b). More recently, such traditional view has been complemented by the recognition that trimeric Gi proteins, such as Gα13 (Gong et al, 2010) and Gαi3 (Leyme et al, 2015), are also implicated in integrin "outside-to-in" signaling. Despite its relevance under pathologic conditions such as cancer invasion and metastatic dissemination, the dynamics of integrin-centered macromolecular complexes in signal transduction remains poorly understood (Cooper & Giancotti, 2019).

Ion channels are increasingly recognized as major partners of integrin-centered macromolecular complexes (Arcangeli & Becchetti, 2006; Qiu et al, 2014; Genova et al, 2017; Becchetti et al, 2019; Gunasekar et al, 2019; Alghanem et al, 2021), and evidence is especially wide for the K+ channel encoded by the human ether-à-go-go-related gene 1 (hERG1) (Arcangeli et al, 2023). In cancer cells, hERG1 is often aberrantly over expressed compared to the normal counterpart, and its expression often increases during the neoplastic progression (Lastraioli et al, 2004, 2012). hERG1 regulates different aspects of cancer cell behavior that depend on cell adhesion to ECM proteins such as fibronectin (FN) and laminin (Becchetti et al, 2022). In particular, hERG1 current ($I_{hERG1}$) amplitude

[1]Department of Experimental and Clinical Medicine, Section of Internal Medicine, University of Florence, Florence, Italy  [2]Department of Physics, University of Florence, and Florence Section of INFN, Florence, Italy  [3]Université d'Orléans and Centre de Biophysique Moléculaire (CBM), CNRS UPR 4301, Orléans, France  [4]Department of Experimental and Clinical Biochemical Sciences, Section of General Pathology, University of Florence, Florence, Italy  [5]Department of Physics, University of Leiden, Leiden, Netherlands  [6]Department of Biotechnology and Biosciences, University of Milano Bicocca, Milan, Italy  [7]Department of Medical Biotechnologies, University of Siena, Siena, Italy  [8]CSDC (Center for the Study of complex dynamics), University of Florence, Florence, Italy

Correspondence: annarosa.arcangeli@unifi.it
*Claudia Duranti and Jessica Iorio contributed equally as first authors
†Andrea Becchetti and Annarosa Arcangeli contributed equally as last authors

generally increases upon integrin-dependent cell adhesion, which leads to hyperpolarization of the resting membrane potential ($V_{rest}$) (Arcangeli et al, 1993; Cherubini et al, 2005; Crociani et al, 2013; Chioccioli Altadonna et al, 2022), which is accompanied by the formation of a hERG1/$\beta$1 integrin macromolecular complex (Cherubini et al, 2005; Crociani et al, 2013; Lastraioli et al, 2015). So far, such complex has been only observed in cancer, where the lack of the canonical accessory subunit KCNE1 allows the channel to associate with $\beta$1 integrins (Becchetti et al, 2017; Lottini et al, 2023). Furthermore, $\beta$1 integrins preferentially recruit hERG1 in the closed conformational state (Becchetti et al, 2017; Petroni et al, 2020). In mouse models, the hERG1/$\beta$1 integrin complex modulates different stages of tumorigenesis, which appear to be modulated by distinct hERG1 conformational states (Becchetti et al, 2019). The open channel affects the FAK/extracellular-regulated kinases (ERKs) pathway, which controls local tumor growth, whereas the integrin-bound closed channel regulates the PI3K/Akt pathway implicated in metastatic dissemination (Becchetti et al, 2017).

Despite the relevance of hERG1 in the pathophysiology of cancer, the reasons of its aberrant expression in tumors and the regulation of the hERG1/$\beta$1 integrin complex dynamics are largely unknown. To fully understand the dynamics of complex biological mechanisms, it is necessary to combine experimental data with comprehensive mathematical models (Klipp & Liebermeister, 2006; Saez-Rodriguez et al, 2015) which also allow to cover some inevitable experimental gaps (Rao & Esposito, 2016; Fang & Wang, 2020). Hence, to fully understand the hERG1 interplay with the $\beta$1 integrin in cancer, we both determined experimentally the molecular signals and developed a mathematical model to quantitatively explain the dynamics of such interaction. By combining experimental data obtained in model cells and pancreatic and colorectal cancer cell lines with a deterministic model, we defined the molecular elements and the kinetic features of a novel signaling pathway by which the hERG1/$\beta$1 integrin complex modulates f-actin organization and hence promotes cell migration.

# Results

### Integrin activation stimulates hERG1 expression and translocation to the plasma membrane

We first studied how integrin engagement stimulates hERG1 current ($I_{hERG1}$) in HEK 293 cells stably expressing hERG1 (HEK–hERG1 cells) seeded on FN-coated dishes, which activates $\beta$1 integrins (Arcangeli et al, 1993; Hofmann et al, 2001). Maximal adhesion to FN and cell spreading was reached by 90 min ($T_{90}$; Fig 1A). At $T_{90}$, $I_{hERG1}$ displayed an approximately fourfold increase in peak absolute value, compared with $T_0$. The effect was accompanied by a resting potential ($V_{rest}$) hyperpolarization of $\approx$20 mV. Subsequently, $I_{hERG1}$ decayed to stable values around –450 pA (Fig 1B), and $V_{rest}$ slowly depolarized to –30 mV (Fig 1C). The time course of $I_{hERG1}$ and $V_{rest}$ closely matched, which is confirmed by the very good correlation between the two parameters ($R^2$ = 0.9347 Pearson's coefficient, $P$ = 0.0003) (Fig 1D). No variations of either $I_{hERG1}$ or $V_{rest}$ occurred in HEK–hERG1 cells seeded on BSA or

polylysine, in agreement with previous reports (Cherubini et al, 2005; Chioccioli Altadonna et al, 2022).

We next studied the molecular bases of $I_{hERG1}$ stimulation. At a fixed time, the average whole-cell $I_{hERG1}$ is given by:

$$I_{hERG1} = \gamma NP_o(V_M - E_K) \tag{1}$$

where $\gamma$ is the single-channel conductance, N is the number of hERG1 channels expressed in the plasma membrane and available to activation, $P_o$ is the probability that an individual channel is open, and $V_m$ and $E_K$ have their usual meaning. For a voltage-dependent channel such as hERG1, $P_o$ is given by the product of the voltage-dependent activation and inactivation parameters (Becchetti et al, 2022). At a certain $V_m$, the whole-cell current can increase because of an alteration of one or more of the $\gamma$, N, and $P_o$ parameters. We first analyzed the hERG1 steady-state activation and inactivation. After cell seeding on FN, scarce changes were observed in the activation (Chioccioli Altadonna et al, 2022) and inactivation $V_{1/2}$ values and the time constant of channel deactivation ($\tau$) at different time points. Fig 1E shows $I_{hERG1}$ traces relative to activation and inactivation protocols, and the corresponding activation and inactivation curves from representative HEK–hERG1 cells, respectively, sampled at $T_5$ and $T_{90}$ on FN. The overall results (current density [$I_{dens}$ pA/pF], deactivation [$\tau_{DEACT}$], inactivation and activation $V_{1/2}$ [Inact $V_{1/2}$ and Act V $_{1/2}$, respectively]) are summarized in Table 1 and Fig S1. Similar results were obtained by activating $\beta$1 integrins with the specific activating antibody TS2/16. A 90 min incubation of BSA-seeded cells with TS2/16 produced $I_{hERG1}$ stimulation and $V_{rest}$ hyperpolarization, with no change in activation $V_{1/2}$ (Fig 1F).

Given that neither cell adhesion on FN nor direct $\beta$1 integrin stimulation modified hERG1 gating; we studied the dynamics of hERG1 expression, by testing *herg1* RNA and hERG1 protein. In cells seeded on FN, *herg1* RNA progressively increased. Its amount more than doubled by $T_{120}$ and remained approximately constant up to $T_{300}$ (Fig 2A). The time course of hERG1 protein expression was quantified by Western blot (WB), which reveals hERG1 protein as a 135 kD band (the core glycosylated protein, present in the ER) and two 150–155 kD bands. These correspond to the fully glycosylated hERG1 protein, which is found in both the plasma membrane and intracellular compartments (i.e., the Golgi apparatus, during hERG1 trafficking towards the plasma membrane, and endosomes, during channel degradation) (Zhou et al, 1998; Guasti et al, 2008). Although the 135 kD band was not altered by cell adhesion on FN (Fig 2B, red circles, indicated as "Lower band"), the intensity of the 150–155 kD bands sharply increased at $T_{30}/T_{60}$, peaked at $T_{90}$, and decreased thereafter (Fig 2B, green circles, indicated as "Upper bands"). Overall, the expression of the fully glycosylated hERG1 protein was more than quadrupled.

We next determined hERG1 localization at the plasma membrane level (or close to it) by immunofluorescence (IF), using the scFv derivative of an anti-hERG1 mAb (Duranti et al, 2018) on gently fixed, unpermeabilized cells, and quantifying the signal measured within the area identified by the white masks highlighted in the pictures in Figs 2C and S2A (details are in the see the Materials and Methods section). Cell adhesion to FN induced a fourfold increase of hERG1 IF signal at $T_{90}$, followed by a progressive decay (Fig 2C, right panel). These results were corroborated by flow cytometry (FC) experiments,

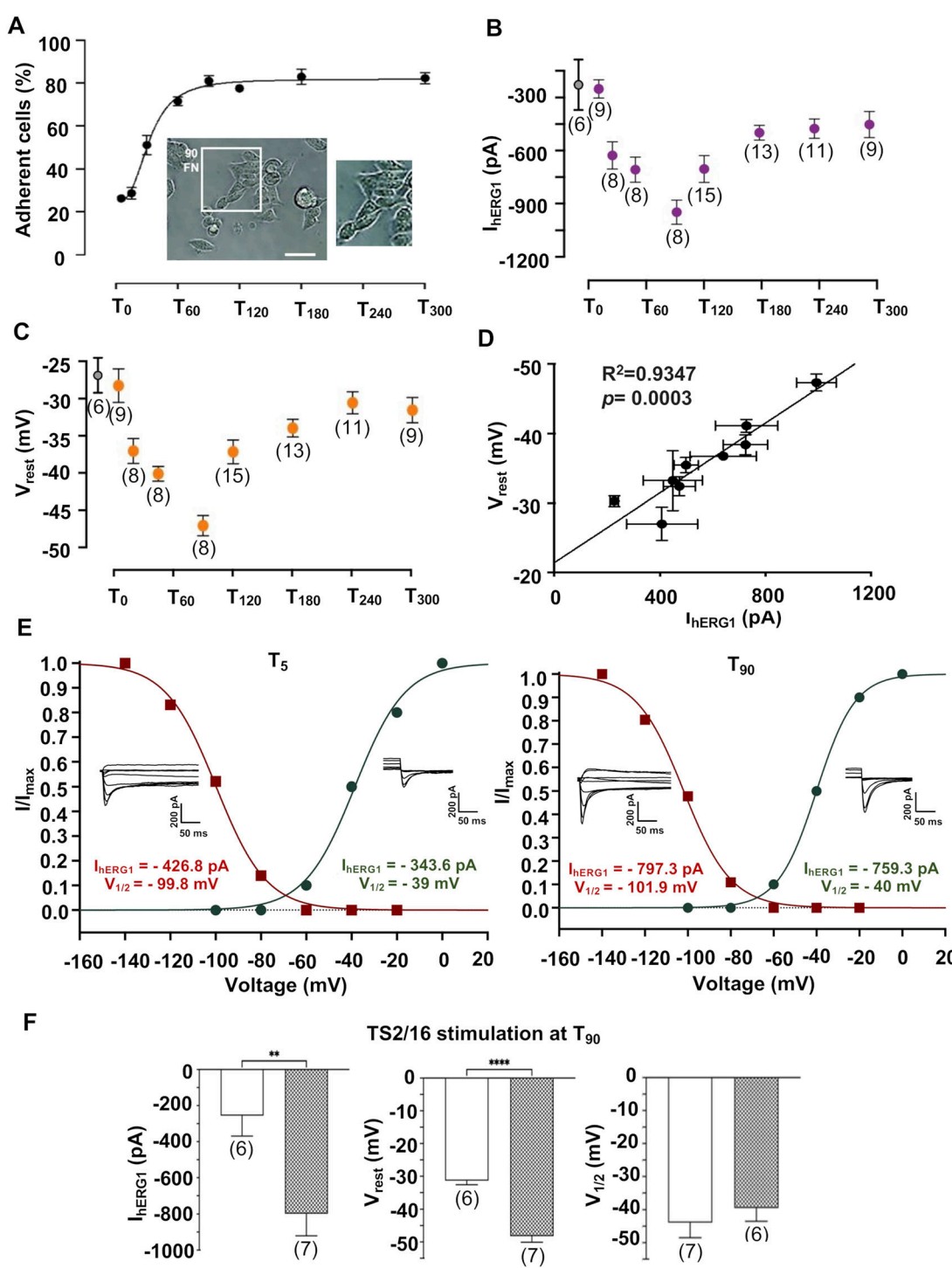

**Figure 1. Dynamics of cell adhesion, hERG1 currents ($I_{hERG1}$), and resting potential ($V_{rest}$) of HEK–hERG1 cells after either adhesion on FN or $\beta$1 integrin stimulation.**
Cells were seeded on FN-coated dishes in serum-free (BSA-containing) medium and monitored from "time zero" ($T_0$, i.e., in cells recovered from a preparatory culture, before seeding) up to 300 min ($T_{300}$). **(A)** Time course of cell adhesion on FN. Representative images are in the insets (n = 3). **(B)** Time course of $I_{hERG1}$ in cells seeded on FN (purple circles); the grey circle refers to $T_0$ data. Data are mean values ± s.e.m. obtained from at least four cell patch clamp experiments. The number of cells analyzed for each time point is shown in brackets. **(C)** Time course of the resting potential ($V_{rest}$) in cells seeded on FN (yellow circles); the grey circle refers to $T_0$ data. Data are mean values ± s.e.m. obtained from at least four cell patch clamp experiments. The number of cells analyzed for each time point is shown in brackets. **(D)** Correlation between $I_{hERG1}$ and $V_{rest}$ in cells seeded on FN (results obtained from at least four cell patch clamp experiments). **(E)** hERG1 activation and inactivation curves of a representative cell at $T_5$ (panel on the left) and of a representative cell at $T_{90}$ (panel on the right) after cell adhesion on FN; the corresponding inward traces are also shown. The fitting confidence intervals of the two curves are: T5: CI 95% ACT: −42.11 to −34.91; CI 95% INACT: −102.0 to −97.67; T90: CI 95% ACT: −40.75 to −39.25; CI 95% INACT: −104.3 to −99.61. **(F)** $I_{hERG1}$, $V_{rest}$, and hERG1 activation $V_{1/2}$ in HEK–hERG1 cells stimulated for 90 min ($T_{90}$) with the $\beta$1 integrin–activating mAb (TS2/16). Data are mean values ± s.e.m. obtained

performed using the full-length anti-hERG1 mAb on live, unper-meabilized cells. This avoids undesired antibody penetration into the cells, hence assuring exclusive labeling of hERG1 in the plasma membrane (Guasti et al, 2008; Petroni et al, 2020). Cell adhesion to FN increased the hERG1 FC signal up to $T_{90}$, which was followed by a slow decay (Fig 2D; the right panel reports the time course of the mean florescence intensity of the area under the curve). Identical results were obtained after $\beta 1$ integrin activation by TS2/16 anti-body (Fig 2E), whereas no effects were observed on $\beta 1$ integrin expression (Fig 2F). The FC plots of each time points were reported in Fig S2B. A clear correlation emerged between FC data, which mirrored hERG1 expression on the plasma membrane and the $I_{hERG1}$ values reported in Fig 1B (Fig 2G; $R^2$ = 0.8010, $P$ = 0.0011). These results show that the integrin-dependent $I_{hERG1}$ stimulation is caused by a larger amount of hERG1 in the plasma membrane, with no substantial alterations of hERG1 gating.

## Integrin activation increases hERG1 protein translocation by engaging trimeric Gαi proteins

We next analyzed how integrins regulate hERG1 translocation to the plasma membrane. Following up our previous observation that the FN-induced hyperpolarization in neuroblastoma cells is inhibited by pertussis toxin (PTX; Arcangeli et al, 1993), we first tested the effect of blocking $G_i$ proteins. In HEK–hERG1 cells treated overnight with PTX, seeding on FN produced no increase in $I_{hERG1}$ (Fig 3A), no $V_{rest}$ hy-perpolarization (Fig 3B), and no increase in hERG1 protein expression on the plasma membrane (Fig 3C) at $T_{90}$. Based on available evidence (Pietruck et al, 1996; Gong et al, 2010; Shen et al, 2012; Leyme et al, 2015), we hypothesized the PTX effect could depend on $G\alpha_{i3}$ subunit blockage. Indeed, $G\alpha_{i3}$ was expressed in HEK–hERG1 cells in cells seeded on BSA, and its expression increased upon cell adhesion to FN (Fig 3D). Furthermore, $G\alpha_{i3}$ co-immunoprecipitated (Fig 3E, left panel) and co-localized with $\beta 1$ integrin (Fig 3E, right panel), reaching the maximal levels at $T_{90}$ (Fig 3E, bar graph on the bottom). The as-sociation with $\beta 1$ integrin was specific, as $G\alpha_{i3}$ did not co-immunoprecipitate with hERG1 (Fig 3F). PTX abolished the $G\alpha_{i3}/\beta 1$ integrin co-immunoprecipitation (Fig 3G, left panel) and co-localization (Fig 3G, right panel). We conclude that PTX prevented the FN-dependent stimulation of hERG1 expression and translocation to the plasma membrane by impairing $G\alpha_{i3}$ interaction with $\beta 1$ integrins.

We further investigated the signaling pathways upstream to $G\alpha_{i3}$, by testing whether the non-receptor guanine nucleotide exchange factor (GEF) girdin was involved (Leyme et al, 2015). In HEK–hERG1 cells seeded on FN, girdin was expressed (Fig 3H, image on the left) and co-localized with $\beta 1$ integrins (Fig 3H, image on the right), with a time course similar to that described for $G\alpha_{i3}$ (Fig 3H, bar graph on the bottom). Silencing girdin with small interfering RNAs (Fig S3) impaired hERG1 translocation to the plasma membrane, as evaluated by FC (Fig 3I) and IF (Fig 3J). Next, we studied the signaling pathway downstream to Gαi3, by blocking either G $\beta\gamma$ subunits recruitment (with gallein) or the subsequent PI3K/Akt pathway (with LY 294002). Both gallein and LY294002 inhibited the FN-dependent translocation of hERG1 to the plasma membrane (Fig 3K),

pointing to the implication of the G$\beta\gamma$ and PI3K/Akt signals. Consis-tently, PTX significantly decreased Akt phosphorylation in HEK–hERG1 cells, at $T_{15}$ on FN (Fig 3L). Thus, $\beta 1$ integrin activation recruits and activates the non-receptor GEF girdin, leading to $G\alpha_{i3}$ association and triggering a PI3K/pAkt-dependent pathway that increases hERG1 protein synthesis and translocation to the plasma membrane.

## The dynamics of hERG1/$\beta 1$ integrin complex formation

To study the kinetics of formation of the macromolecular complex, we monitored the time course of (i) hERG1/$\beta 1$ co-immunoprecipitation (Figs 4A and S4) and (ii) the IF signal obtained by labeling the cells with a single-chain diabody which selectively recognizes the hERG1/$\beta 1$ integrin complex (scDb–hERG1–$\beta 1$; Duranti et al, 2021b) (Fig 4B, left panel). The signal selectively present at the plasma membrane level was quantified as in Fig 2C (lower panels of Fig 4B). Data were collected from $T_0$ to $T_{300}$, after cell seeding on FN. The hERG1/$\beta 1$ complex amount increased approximately four times from $T_0$ to $T_{90}$, and progressively decayed afterwards (Fig 4A and B green symbols). A good correlation was observed between the results of the two methods ($R^2$ = 0.8536, $P$ = 0.0004, Fig 4C). No complex formation was observed in cells seeded on BSA (Fig 4A, white circles), or in the cytoplasm of FN-seeded cells (Fig 4B, red symbols). After $T_{90}$–$T_{120}$, the expression of both the channel (see Fig 2) and the hERG1/$\beta 1$ complex (Fig 4A and B) tended to de-crease. We asked whether such decrease was caused by hERG1 degradation through endosomes (Foo et al, 2016), by studying the co-localization of either hERG1 (Fig 4D), the hERG1/$\beta 1$ complex (Fig 4E), or the $\beta 1$ integrin (Fig 4F) with RAB5, a marker of early endosomes (Gorvel et al, 1991). hERG1 was labeled with our hERG1-specific scFv (Duranti et al, 2018), whereas the hERG1/$\beta 1$ complex was labeled with a fluorescent scDb–hERG1–$\beta 1$ (Duranti et al, 2021a). In Fig 4D and E, the merged pictures are shown (all the pictures are in Fig S5). In these determinations, we selected the cytoplasmic IF signals, as explained in see the Materials and Methods section and the figure legend. HERG1 co-localized with RAB5 (i.e., it was present in early endosomes) at early stages of cell seeding on FN. Co-localization progressively decreased up to $T_{90}$ (when hERG1 expression on the plasma membrane was maximal) and increased again afterwards. At $T_{120}$, hERG1 expression had returned to the initial level (Fig 4D, see the Manders' Overlap coefficient in the right panel). On the contrary, the hERG1/$\beta 1$ integrin complex was virtually absent from endosomes throughout the ex-periment (Fig 4E, see the Manders' Overlap coefficient in the right panel). No time-dependent variation in the co-localization of RAB5 with $\beta 1$ integrin was observed (Fig 4F, see the Manders' Overlap co-efficient in the right panel). Overall, data in Fig 4 suggest that the biphasic expression of the hERG1/$\beta 1$ integrin complex depends on the hERG1 availability in the plasma membrane.

## The hERG1/$\beta 1$ integrin complex regulates hERG1 localization to the plasma membrane

The above conclusion was supported by the demonstration that hampers hERG1 translocation by PTX, gallein, or LY294002 treatment

---

from at least four cell patch clamp experiments. The number of cells analyzed for each time point is shown in brackets. Time zero is defined as the timepoint corresponding to the cells seeding. *$P$ < 0.05; **$P$ < 0.01, and ***$P$ < 0.001. The original data relative to this figure are shown in Fig S1.

**Table 1. hERG1 current densities (Idens), time constants of deactivation (taudeact), and V$_{1/2}$ of inactivation (Inact V$_{1/2}$) and activation (Act V$_{1/2}$), at the indicated times (first column) after cell seeding on FN.**

| Time (min) | I$_{dens}$ (pA/pF) | | | τ$_{DEACT}$ (ms) | | | Inact V$_{1/2}$ (mV) | | | Act V$_{1/2}$ (mV) | | |
|---|---|---|---|---|---|---|---|---|---|---|---|---|
| | Mean | s.e.m | n | Mean | s.e.m | n | Mean | s.e.m | n | Mean | s.e.m | n |
| 5 | 31.3 | 2.8 | 40 | 53.5 | 3.4 | 32 | 98.7 | 1.5 | 18 | 42.3 | 1.3 | 31 |
| 45 | 65.3 | 5.7 | 31 | 52.1 | 3.1 | 22 | 100.6 | 1.0 | 18 | 41.9 | 1.3 | 23 |
| 90 | 84.4 | 8.2 | 29 | 51.1 | 1.8 | 21 | 101.5 | 1.7 | 13 | 42.0 | 1.4 | 20 |
| 120 | 53.3 | 8.2 | 8 | 45.9 | 5.8 | 7 | 102.4 | 0.9 | 4 | 43.2 | 1.4 | 6 |
| 180 | 46.4 | 5.9 | 17 | 49.1 | 3.3 | 17 | 99.7 | 0.9 | 9 | 40.5 | 1.3 | 14 |
| 240 | 57.4 | 8.0 | 14 | 48.6 | 2.4 | 14 | 100.9 | 0.6 | 9 | 38.4 | 1.2 | 18 |
| 300 | 53.4 | 5.9 | 24 | 48.1 | 3.3 | 18 | 101.5 | 1.2 | 9 | 42.1 | 1.4 | 17 |

Data are mean values ± s.e.m. obtained from the number of patched cells reported in the column labeled as "n," obtained in at least four different experiments.

(see Fig 3) strongly impaired the hERG1/β1 integrin complex formation, in cells adhering to FN (Fig 5A). Collectively, our results show that the macromolecular complex assembly requires newly expressed hERG1 channels, without relying on the interaction between β1 integrins and the unbound channels already located in the plasma membrane, despite the numerosity of the latter (see Table 2 and Supplemental Data 1). This notion points to a specific regulatory interplay between the hERG1/β1 integrin complex and the channel translocation mechanism, which we tested by studying whether impairing the complex formation in the plasma membrane–affected hERG1 translocation to the plasma membrane. First, we partly substituted β1 integrin with the canonical hERG1 accessory subunit KCNE1, which prevents the complex formation (Becchetti et al, 2017). To this aim, we transiently transfected with hERG1–GFP (i) wild-type HEK cells and (ii) cells stably expressing KCNE1 (HEK–KCNE1), to, respectively, obtain HEK–hERG1–Tr cells and HEK–KCNE1–hERG1–Tr cells. Both cell types expressed a functional hERG1 (Fig S5). When seeded on FN, HEK–KCNE1–hERG1–Tr displayed a lower I$_{hERG1}$ increase (1.5-folds) at T$_{90}$ (Fig 5B), a smaller amount of the hERG1/β1 integrin complex (Fig 5C), and a less hERG1 translocation to the plasma membrane (Fig 5D). These results confirm that β1 integrin engagement modulates I$_{hERG1}$ by stimulating channel trafficking and suggest that the physical link between hERG1 and β1 integrin causes a positive feedback for channel translocation. In other words, when the complex formation is hampered, β1 integrin activation is not sufficient to sustain hERG1 translocation. Because, however, we could not exclude that the KCNE1/hERG1 association could directly inhibit channel translocation, we further tested our hypothesis with two independent approaches. First, we relied on our previous observation that β1 integrins preferentially recruit hERG1 channels in the closed state (Becchetti et al, 2017), as indicated by the study of HEK cells transfected with the mutant constructs hERG1–K525C or hERG1–R531C. These mutants, at physiological V$_{rest}$, preferentially reside in the open (hERG1–K525C) or in the closed (hERG1–R531C) state (Zhang et al, 2004). Therefore, hERG1–R531C forms normal complexes with β1 integrins, whereas hERG1–K525C does not. After seeding on FN, *herg1* mRNA expression increased in HEK–hERG1–R531C cells as it did in HEK–hERG1 cells (Fig 2A), but not in HEK–hERG1–K525C cells (Fig 5E). This suggests a negative feedback depending on the amount of hERG1 current on mRNA expression,

which is not further investigated. Furthermore, although hERG1–K525C scarcely reached the plasma membrane, hERG1–R531C was constitutively present, as witnessed by FC (Fig 5F). Consistently, the hERG1/β1 integrin complex (determined by co-IP) never occurred in HEK–hERG1–K525C cells, whereas it was normally assembled in HEK–hERG1–R531C cells (Fig 5G). Fig 5F and G show the densitometric data; the original FCs and co-IPs are in Fig S5. These data were confirmed by IF with the fluorescent scDb–hERG1–β1 (Fig 5H). Furthermore, co-localization data with RAB5 show that hERG1–K525C was mainly found in endosomes, whereas hERG1–R531C was not (Fig 5I). As a second approach, we impaired the hERG1/β1 integrin complex formation by treating live HEK–hERG1 cells seeded on FN with scDb–hERG1–β1. In agreement with the above results, this treatment inhibited complex formation (Fig 5J) and hERG1 translocation to the plasma membrane at T$_{90}$ (Fig 5K). We conclude that assembly of the hERG1/β1 integrin complex drives the hERG1 expression/translocation process.

### The dynamics of hERG1 interaction with integrins in cancer cells

We then studied whether the above mechanisms were also operant in cancer cells, where hERG1 is constitutively overexpressed and the hERG1/β1 integrin complex is present (Cherubini et al, 2005; Crociani et al, 2013; Lastraioli et al, 2015, see also Figs 6E and S6). We analyzed a pancreatic ductal adenocarcinoma cell line (PANC-1) and a colorectal cancer cell line (HCT116), showing that (i) cells adhered to FN, attaining the plateau value and spreading on the substrate at T$_{90}$ (Fig 6A); (ii) the average V$_{rest}$ of FN-seeded cells hyperpolarized, between T$_0$ and T$_{90}$, by ~20 mV in PANC-1 and 10 mV in HCT116 (Fig 6B). As in HEK–hERG1 cells, PANC-1 and HCT116 cell adhesion on FN-stimulated hERG1 localization in the plasma membrane, with a peak at T$_{90}$, which was followed by a comparatively slower decrease (Fig 6C). A similar time course was observed in IF experiments with the anti-hERG1 scFv (Fig 6D). The hERG1/β1 complex formation, determined by either co-IP experiments (Fig 6E) or IF with fluorescent scDb–hERG1–β1 (Fig 6F), also showed a progressive increase in both cell types seeded on FN up to T$_{90}$, followed by a slower decay. The kinetics of these processes is quantified by the mathematical model described later.

Finally, we tested if the molecular mechanism that regulates hERG1 translocation and the hERG1/β1 integrin complex formation

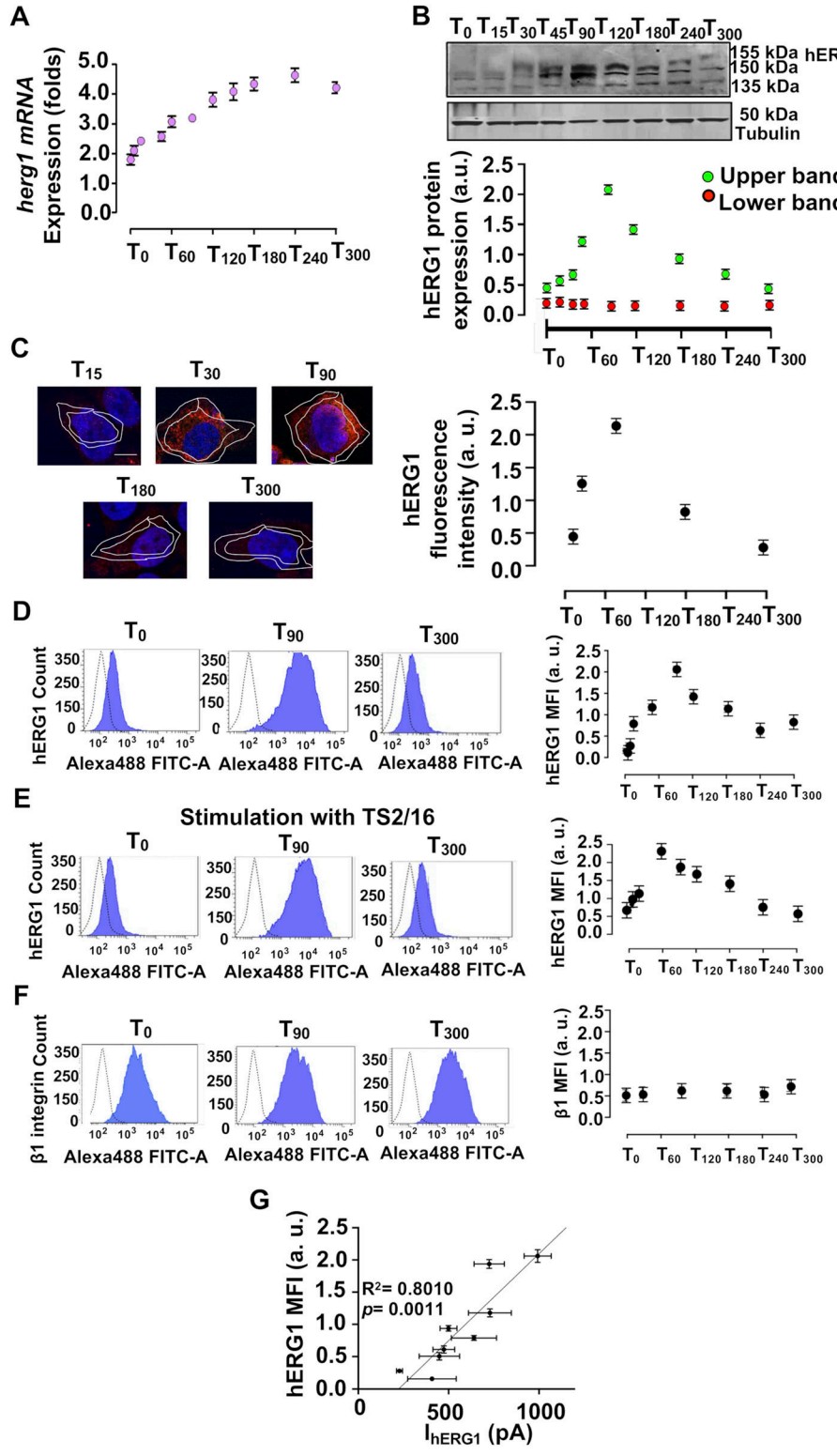

**Figure 2. Dynamics of FN- and integrin-dependent hERG1 expression and translocation to the plasma membrane in HEK–hERG1 cells.**

**(A)** Time course of *herg1* mRNA expression in HEK–hERG1 cells seeded on FN. Data, reported as 2-DCt, are mean values ± s.e.m. (n = 3). **(B)** Time course of hERG1 protein expression determined by Western blot (WB) in HEK–hERG1 cells seeded on FN. Representative WB (upper panel). Densitometric analysis of hERG1 protein expression considering either the upper bands (155 kD and 150 kD, green circles) or the lower band (135 kD, red circles) (lower panel). Data are mean values ± s.e.m obtained in three independent experiments. **(C)** Time course of hERG1 protein expression determined by IF in HEK–hERG1 cells seeded on FN. Cells were stained with the scFv–hERG1 (Duranti et al, 2018). Representative images (scale bar 100 $\mu$m) on the left and quantification on the right. The quantification was performed considering only the membrane signal, highlighted by the white masks (shown in the pictures) drawn as detailed in Materials and Methods section. For each condition, the fluorescence relative to 20 cells was analyzed. Data have been normalized on the zero value and reported as fold increase arbitrary units (a. u.) on a scale ranging between 0 and 2.5. **(D)** Time course of hERG1 protein expression determined by flow cytometry (FC) in HEK–hERG1 cells seeded on FN. Representative FC plots at $T_0$, $T_{90}$, and $T_{300}$, representative of three independent experiments (panels on the left). Time courses of mean fluorescent intensity. Values are expressed as mean fluorescence intensity of the area under the curve (MFI) (panel on the right). Data have been normalized on the zero value and reported as fold increase arbitrary units (a. u.) on a scale ranging between 0 and 2.5. Data are mean values ± s.e.m. (n = 3). **(E)** Time course of hERG1 protein expression determined by FC in HEK–hERG1 cells seeded on BSA and stimulated with the $\beta$1 integrins–stimulating antibody TS2/16. Representative FC plots at $T_0$, $T_{90}$, and $T_{300}$, representative of three independent experiments (panels on the left). Time courses of the mean fluorescent intensity of the area under the curve (MFI) (panel on the right). Data have been normalized on the zero value and reported as fold increase arbitrary units (a. u.) on a scale ranging between 0 and 2.5. Data are mean values ± s.e.m. (n = 3). **(F)** Time course of $\beta$1 integrin expression determined by FC in HEK–hERG1 cells seeded on FN. Representative FC plots at $T_0$, $T_{90}$, and $T_{300}$, representative of three independent experiments (panels on the left). Time courses of mean fluorescent intensity of the area under the curve (MFI) (panel on the right). Data have been normalized on the zero value and reported as fold increase arbitrary units (a. u.) on a scale ranging between 0 and 2.5. Data are mean values ± s.e.m. (n = 3). **(G)** Correlation between hERG1 plasma membrane expression (determined as MFI) and $I_{hERG1}$ in cells seeded on FN (results obtained from at least four cell patch clamp experiments). Data are mean values ± s.e.m. (n = 3). *$P < 0.05$; **$P < 0.01$, and ***$P < 0.001$. Fig 2 shows representative data, the whole dataset is displayed in Fig S2.

Source data are available for this figure.

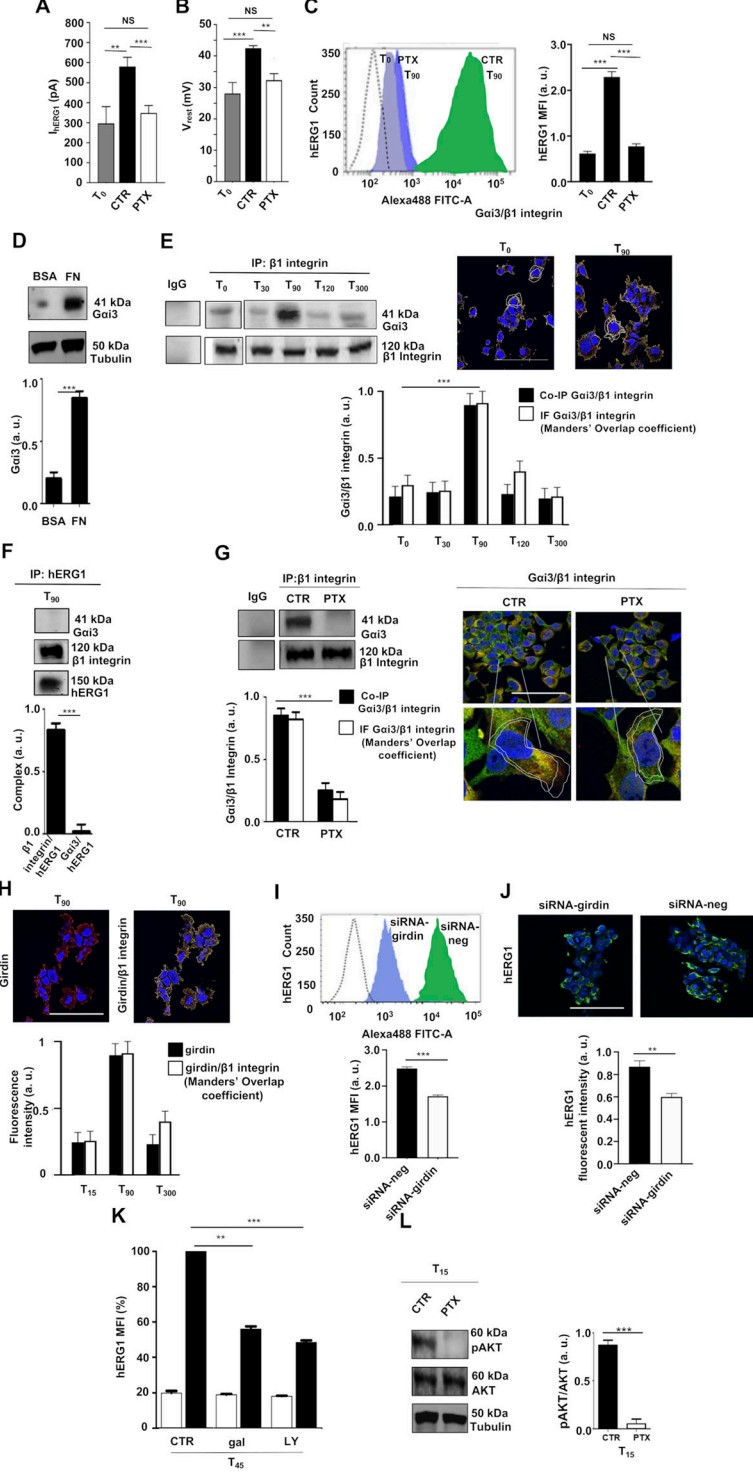

**Figure 3. Involvement of girdin, Gαi3, and PI3K/Akt in the integrin-dependent increase in hERG1 expression and translocation to the plasma membrane in HEK–hERG1 cells.**
**(A, B)** $I_{hERG1}$ and (B) $V_{rest}$ in HEK–hERG1 cells untreated (CTR) or treated overnight with PTX (100 μg/ml, labeled as "PTX") and seeded on FN for 90 min ($T_{90}$). **(C)** hERG1 translocation on the plasma membrane in HEK–hERG1 cells untreated (CTR) or treated overnight with PTX (100 ng/ml final concentration, labeled as "PTX") and seeded on FN for 90 min ($T_{90}$). Representative FC plots of hERG1 expression at $T_0$, $T_{90}$ in HEK–hERG1 cells untreated (CTR, green plot) or treated overnight with PTX (PTX, blue plot) are in the left panel, the hERG1 mean fluorescent intensity of the area under the curve (MFI) is in the right panel. Data relative to untreated cells at $T_0$ are also shown. Data have been normalized on the zero value and reported as fold increase arbitrary units (a. u.) on a scale ranging between 0 and 3.0. Data are mean values ± s.e.m. (n = 3). **(D)** Representative WB (top) and densitometric analysis (bottom) of Gαi3 expression in HEK–hERG1 cells seeded on BSA and FN at $T_{90}$. **(E)** Left panel: representative WBs of co-IP between Gαi3 and the β1 integrin and between Gαi3 and hERG1 in HEK–hERG1 cells seeded on FN at the time points indicated in the figure (total lysates inputs are reported in Fig S3); Right panel: IF representative images (scale bar: 100 μm) of the co-localization of Gαi3 (green signal) and β1 integrin (red signal) in HEK–hERG1 cells seeded on FN at $T_{90}$. The panel on the bottom shows the densiometric analysis of both co-IP and IF data. For each condition, the fluorescence relative to 20 cells was analyzed. (n = 3). a.u. = arbitrary units. **(F)** Representative WBs (top) and densitometric analysis (bottom) of co-IP between Gαi3 and hERG1 in HEK–hERG1 cells seeded on FN at $T_{90}$ (total lysates, indicated as "inputs," are reported in Fig S3). **(G)** Left panel: representative WB of the co-IP of Gαi3 and β1 integrin in HEK–hERG1 cells untreated (CTR) or treated overnight with PTX (PTX) and seeded on FN for 90 min (n = 3); right panel: representative IF images (scale bar: 100 μm) of the co-localization between β1 integrin and Gαi3 in HEK–hERG1 cells untreated (CTR) or treated overnight with PTX (PTX) and seeded on FN for 90 min. The panel on the bottom shows the densiometric analysis of both co-IP and IF data. For each condition, the fluorescence relative to 20 cells was analyzed. Data are mean values ± s.e.m. (n = 3). a.u. = arbitrary units. **(H)** Representative IF images (scale bar: 100 μm) of girdin expression in HEK–hERG1 cells seeded on FN at $T_{90}$ (image on the left) and of the co-localization of girdin (green signal) and β1 integrin (red signal) in HEK–hERG1 cells seeded on FN at $T_{90}$ (image on the right). The panel on the bottom shows the densiometric analysis of both IF and co-IP data at $T_{15}$, $T_{90}$, and $T_{300}$. For each condition, the fluorescence relative to 20 cells was analyzed as detailed in Materials and Methods section. Data are mean values ± s.e.m. (n = 3). a.u. = arbitrary units. **(I)** Effects of siRNA-girding on hERG1 expression of HEK–hERG1 cells. Representative FC plot (top) and densitometric analysis (bottom) of hERG1 expression in HEK–hERG1 cells stimulated with the β1 integrin–activating mAb TS2/16 at $T_{90}$ in the presence of siRNA–girdin (blue plot) and siRNA negative (green control, as controls). The dotted FC plots represent traces at $T_0$. Data have been normalized on the zero value and reported as fold increase arbitrary units (a. u.) on a scale ranging between 0 and 3.0. Data are mean values ± s.e.m. (n = 3); a.u., arbitrary units. **(J)** Effects of siRNA-girding on hERG1 expression of HEK–hERG1 cells. Representative IF images (top) (scale bar: 100 μm) and densitometric analysis (bottom) of hERG1 expression in HEK–hERG1 cells seeded on FN at $T_{90}$ in the presence of siRNA–girdin and siRNA negative (as controls). For each condition, the fluorescence relative to 20 cells was analyzed as detailed in Materials and Methods section. Data are mean values ± s.e.m. (n = 3). a.u., arbitrary units. **(K)** hERG1 MFI in HEK–hERG1 cells after stimulation with the β1 integrin–activating mAb TS2/16 and treatment with gallein or LY294002 at $T_{45}$. Data are mean values ± s.e.m. (n = 3).

**(L)** Representative blot (left) and densitometric analysis (right) of phospho-Akt levels in HEK–hERG1 cells untreated (CTR) or treated overnight with PTX (PTX) and seeded on FN at $T_{15}$. Data are presented as mean values ± s.e.m. (n = 3). a.u., arbitrary units. Membranes were probed with anti-pAkt Thr308 and anti-Akt Thr308 antibodies. NS, not significantly different, **$P < 0.01$, and ***$P < 0.001$. Fig 3 shows representative data, the whole set of data is in Fig S3. Source data are available for this figure.

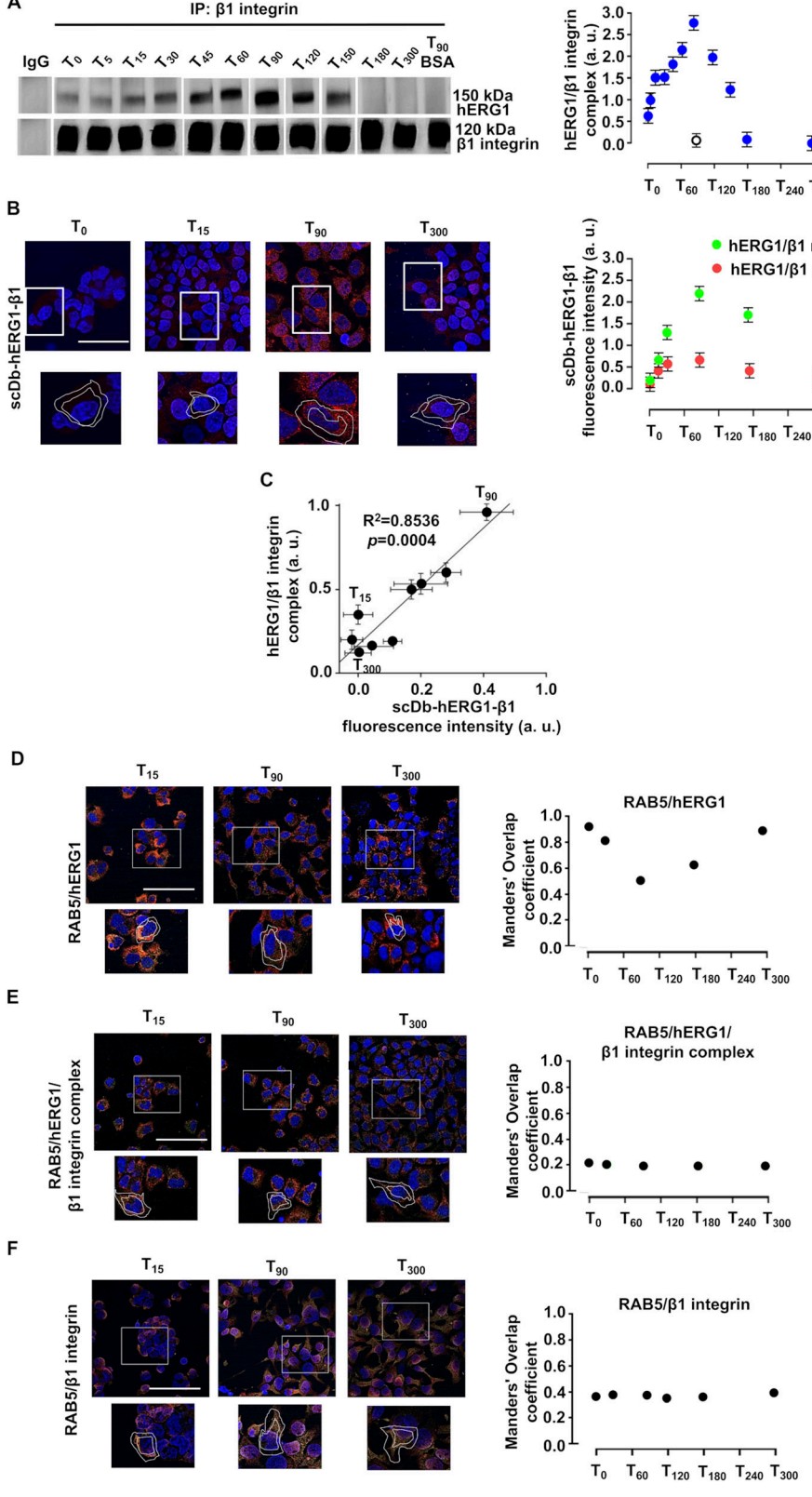

**Figure 4. Dynamics of hERG1/β1 integrin complex in HEK–hERG1 cells seeded on FN.**
**(A)** Time course of the co-IP between hERG1 and β1 integrin in cells seeded on FN or BSA-coated dishes. Left panel: representative experiment; right panel: densitometric analysis (n = 3). Data are reported as fold increase in a. u. (arbitrary unit) between 0 and 3.0 (total lysates, indicated as "inputs" are reported in Fig S4). **(B)** Time course of hERG1/β1 integrin complex expression determined by IF in HEK–hERG1 cells seeded on FN. Cells were stained with the scDb–hERG1–β1 (Duranti et al, 2021a, 2021b). Representative images (scale bar: 100 μm) are on the left and relative quantification on the right. The quantification was performed considering only the membrane signal, highlighted by the white masks shown in the pictures (green circles). In the graph was reported also the cytoplasmatic IF signal (red circles). For each condition, the fluorescence relative to 20 cells was analyzed. Data have been reported as fold increase in a. u. (arbitrary unit) between 0 and 3.0 (n = 3). **(C)** Correlation between hERG1/β1 co-IP and the IF signal obtained by labeling the cells with scDb–hERG1–β1. Data are presented as mean values ± s.e.m. **(D)** Representative IF images (scale bar: 100 μm) of the co-localization of hERG1 (red signal) and RAB5 (green signal) in HEK–hERG1 cells at different time points. Manders' Overlap coefficient reporting hERG1/Rab5 correlation is reported in the right panel. **(E)** Representative IF images (scale bar: 100 μm) of the co-localization of hERG1/β1 integrin complex (red signal) in HEK–hERG1 cells at different time points. Manders' Overlap coefficient reporting RAB5/hERG1/β1 integrin correlation is reported in the right panel. **(F)** Representative IF images (scale bar: 100 μm) of the co-localization of hERG1/β1 integrin in HEK–hERG1 cells at different time points. Manders' Overlap coefficient reporting RAB5/β1 integrin correlation is reported in the right panel. For each condition, the fluorescence relative to 20 cells was analyzed. Data are presented as mean values ± s.e.m. (n = 3). Fig 4 shows representative data, the whole set of data is in Fig S4.
Source data are available for this figure.

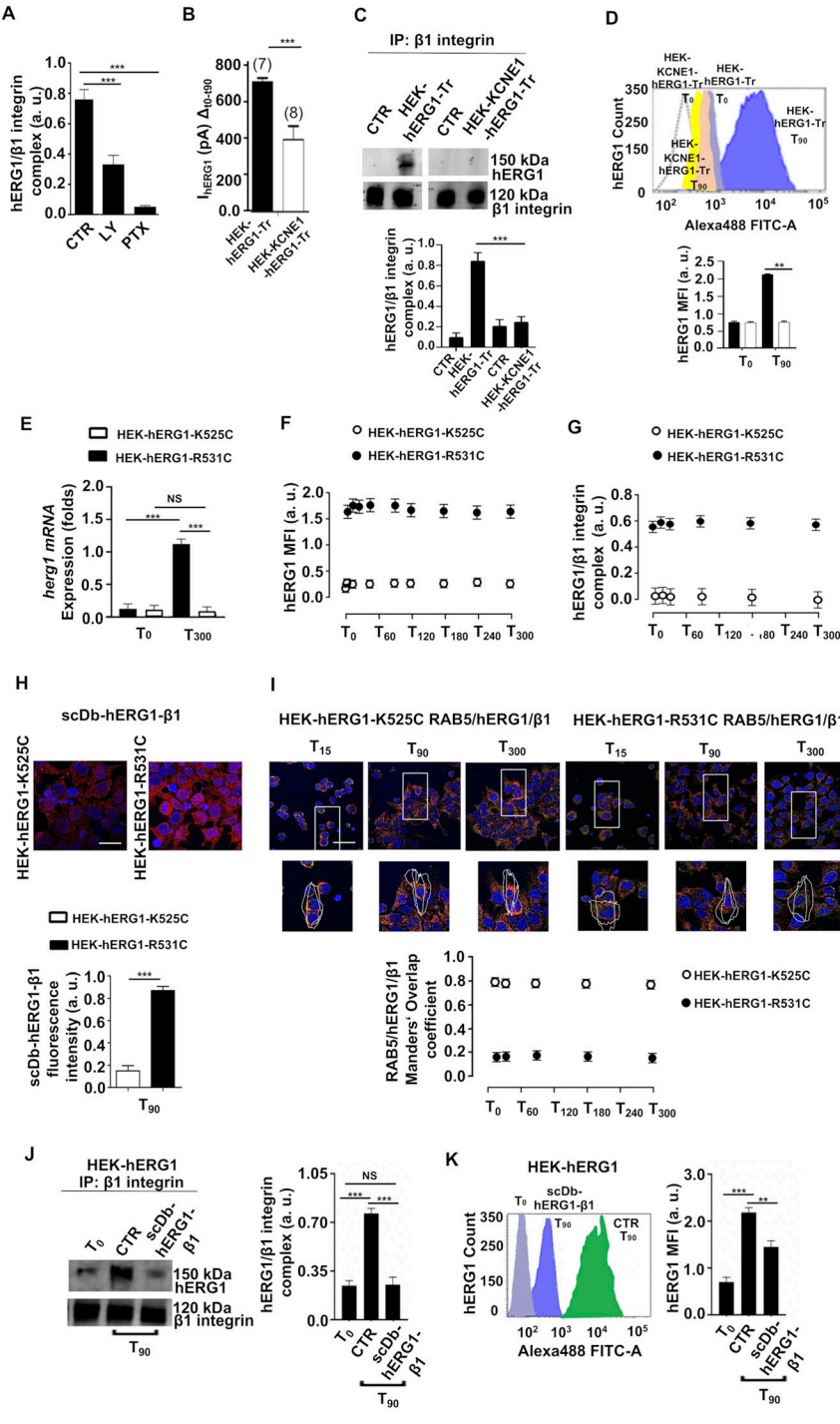

**Figure 5. Dynamics of hERG1/β1 integrin complex in HEK–hERG1 cells transfected either with KCNE1 or with K525C or R531C mutants.**

**(A)** Densitometric quantification of the co-IP between hERG1 and β1 integrin in HEK–hERG1 cells stimulated with TS2/16 and treated with LY294002 and PTX at $T_{90}$. Data are presented as mean values ± s.e.m. ($n$ = 3). a.u., arbitrary units. **(B, C, D)** Effects of KCNE1 overexpression in HEK and HEK–hERG1–Tr cells. **(B)** $I_{hERG1}$ increment (from $T_0$ to $T_{90}$) in HEK cells transiently transfected with hERG1–GFP (HEK–hERG1–Tr, black bar) and in HEK cells stably expressing KCNE1 (HEK–KCNE1) and transiently transfected with hERG1–GFP (HEK–KCNE1–hERG1–Tr, white bar) seeded on FN ($n$ = 3). The number of cells analyzed is shown in brackets. **(C)** Representative blot (top) and corresponding densitometric analysis (bottom) of the co-IP between hERG1 and β1 integrin (performed using the TS2/16 antibody) in HEK–hERG1–Tr and HEK–KCNE1–hERG1–Tr cells seeded on FN at $T_{90}$. Data are presented ad mean values ± s.e.m. ($n$ = 3). a.u. = arbitrary units. **(D)** Representative FC plots (top) of hERG1 expression at $T_0$, $T_{90}$ in HEK–hERG1–Tr and HEK–KCNE1–hERG1–Tr cells seeded on FN. MFI quantification in HEK–hERG1–Tr (black bars) and in HEK–KCNE1–hERG1–Tr cells (white bars) is reported in the bottom panel. Data have been normalized on the zero value and reported as fold increase arbitrary units (a. u.) on a scale ranging between 0 and 2.5. Data are mean values ± s.e.m. ($n$ = 3). **(E)** *herg1* mRNA expression in HEK–hERG1–K525C (white bars) and HEK–hERG1–R531C (black bars) cells seeded on FN, at $T_0$ and $T_{300}$. Data, reported as 2-DCt, are mean values ± s.e.m. ($n$ = 3). **(F)** Time course of the hERG1 mean fluorescence intensity or the area under the curve (MFI) in HEK–hERG1–K525C (white circles) and HEK–hERG1–R531C cells (black circles) seeded on FN. Data have been normalized on the zero value and reported as fold increase arbitrary units (a. u.) on a scale ranging between 0 and 2.0. Original FC plots are in Fig S5. **(G)** Densitometric quantification of the time course of the co-IP between hERG1 and β1 integrin (performed using the TS2/16 antibody) in HEK–hERG1–K525C (white circles) and HEK–hERG1–R531C cells (black circles). Data are mean values ± s.e.m. ($n$ = 3). a.u. = arbitrary units. The original co-Ips blots are in Fig S5. **(H)** Representative IF images (top) (scale bar: 100 $\mu$m) and quantification of fluorescence intensity (bottom) of HEK–hERG1–K525C and HEK–hERG1–R531C cells seeded on FN at $T_{90}$ and stained with scDb–hERG1–β1. For each condition, the fluorescence relative to 20 cells was analyzed. Data are presented as mean values ± s.e.m. ($n$ = 3). **(I)** Representative IF images (scale bar: 100 $\mu$m) of RAB5/β1 co-expression in HEK–hERG1–K525C and HEK–hERG1–R531C cells (top panel). Mander's Overlapping coefficient of RAB5/hERG1/β1 co-localization in HEK–hERG1–K525C (white circles) and HEK–hERG1–R531C cells (black circles) (bottom panel). For each condition, the fluorescence relative to 20 cells was analyzed. Data are presented as mean values ± s.e.m. ($n$ = 3). **(J)** Effects of the treatment with scDb–hERG1–β1 (20 $\mu$g/ml) added to HEK–hERG1 live cells seeded for different times onto FN on hERG1/β1 integrin complex formation. A representative blot is in the left panel, the densitometric data are in the right panel, (total lysates inputs are reported in Fig S5). Data are presented as mean values ± s.e.m. ($n$ = 3). **(K)** Effects of the treatment with scDb–hERG1–β1 (20 $\mu$g/ml) added to live HEK–hERG1 cells seeded on FN at $T_{90}$ on hERG1 membrane translocation. Representative FC plots are in the left panel, the hERG1 MFI is in the right panel. Data have been normalized on the zero value and reported as fold increase arbitrary units (a. u.) on a scale ranging between 0 and 3.0. Data are mean values ± s.e.m. ($n$ = 3). a.u. = arbitrary units. NS, not significantly different, **$P < 0.01$, and ***$P < 0.001$. Fig 5 shows representative data, the whole set of data is in Fig S5.

Source data are available for this figure.

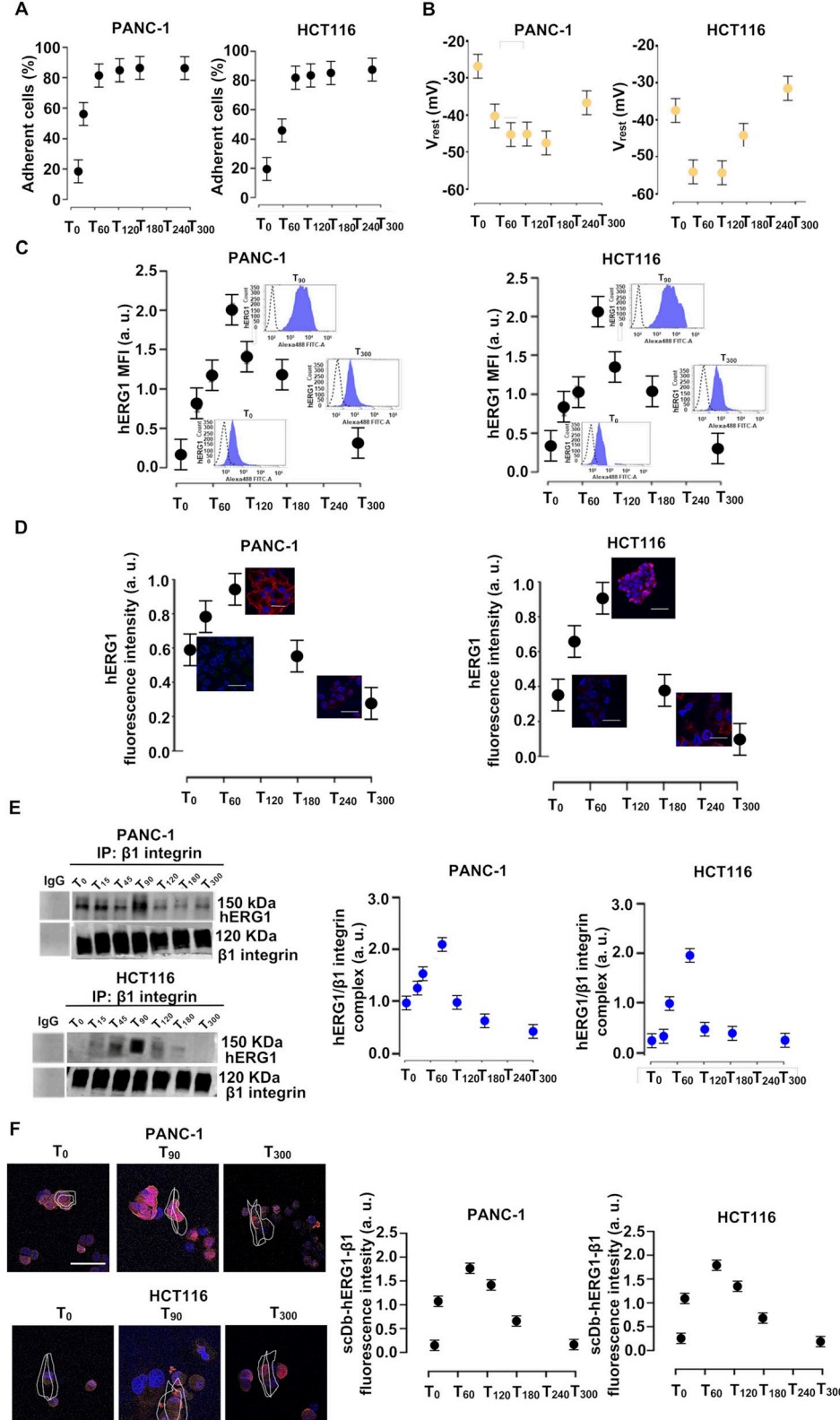

**Figure 6. Dynamics of cell adhesion, $V_{rest}$, hERG1 expression, and hERG1/$\beta$1 integrin complex in PANC-1 and HCT116 cancer cells.**
**(A)** Time course of cell adhesion of PANC-1 and HCT116 cells seeded on FN. Data, expressed as percentage of adherent cells, are mean values ± s.e.m. ($n$ = 3). **(B)** Time course of $V_{rest}$ of PANC-1 and HCT116 cells seeded on FN. At least five cells for each data point were analyzed. Data are presented ad mean values ± s.e.m. ($n$ = 3). **(C)** Time course of hERG1 expression on the plasma membrane in PANC-1 and HCT116 cells seeded on FN, assessed by flow cytometry and expressed as mean fluorescence intensity. Data have been normalized on the zero value and reported as fold increase arbitrary units (a. u.) on a scale ranging between 0 and 2.5. Data are presented as mean values ± s.e.m. ($n$ = 3). **(D)** Time course of hERG1 expression on the plasma membrane in PANC-1 and HCT116 cells seeded on FN, assessed by IF after staining with scFv–hERG1. For each condition, the fluorescence relative to 20 cells was analyzed. Data are presented ad mean values ± s.e.m. ($n$ = 3). **(E)** Time course of hERG1/$\beta$1 integrin complex formation in PANC-1 and HCT116 cells seeded on FN, assessed by co-IP (performed using the TS2/16 antibody). Data have been normalized on the zero value and reported as fold increase arbitrary units (a. u.) on a scale ranging between 0 and 3.0. Data are presented ad mean values ± s.e.m. ($n$ = 3). **(F)** Time course of hERG1/$\beta$1 integrin complex formation in PANC-1 and HCT116 cells seeded on FN, assessed by IF using the scDb–hERG1/$\beta$1. Representative images (scale bar 100 $\mu$m) on the left and quantification on the right. Data have been normalized on the zero value and reported as fold increase arbitrary units (a. u.) on a scale ranging between 0 and 2.5. For each condition, the fluorescence relative to 20 cells was analyzed. Data are presented ad mean values ± s.e.m. ($n$ = 3). Fig 6 shows representative data, the whole set of data is in Fig S6.
Source data are available for this figure.

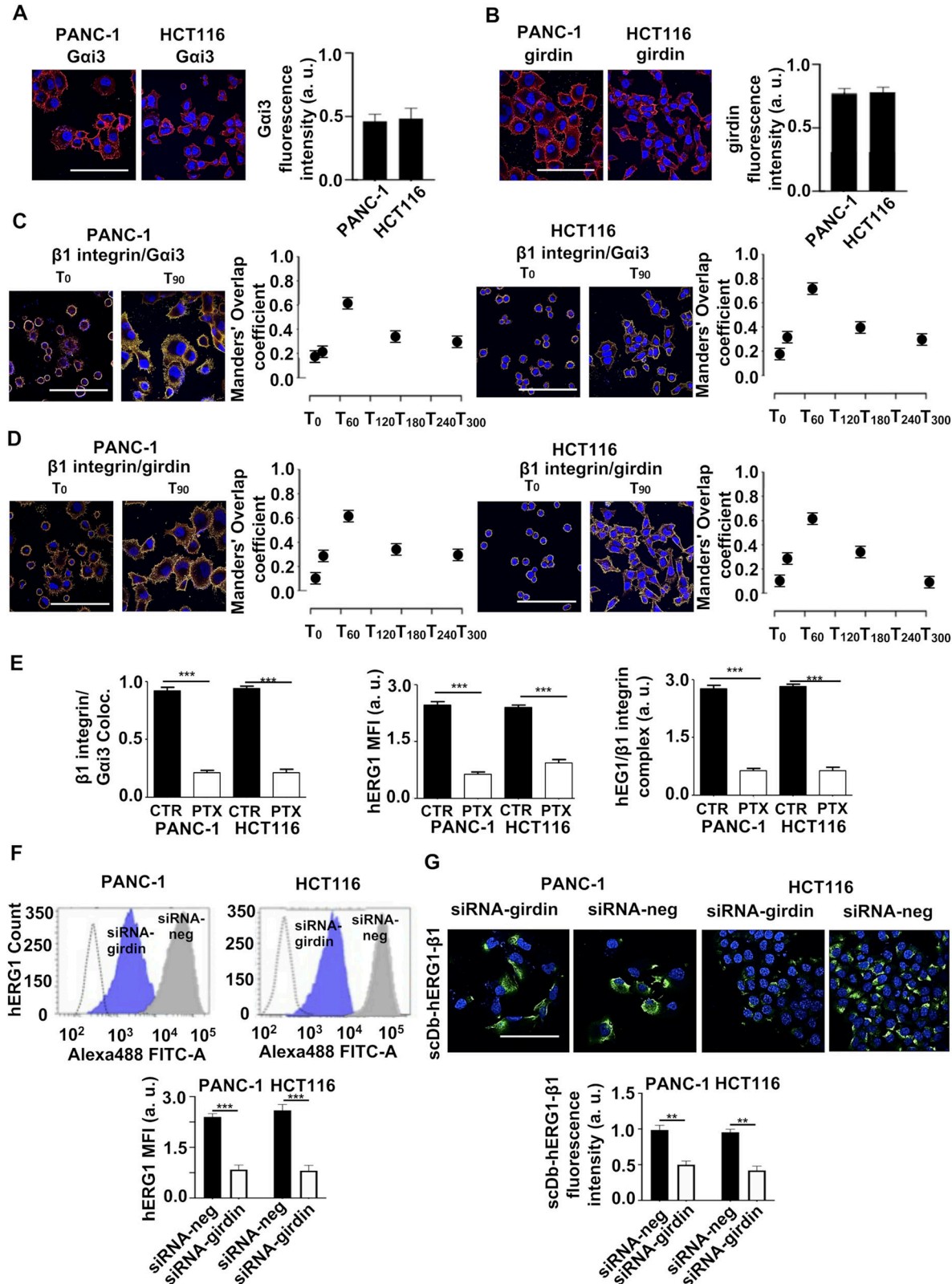

**Figure 7. Involvement of girdin and Gαi3 in the integrin-dependent increase in hERG1 channel translocation on the plasma membrane and hERG1/β1 integrin complex formation in PANC-1 and HCT116 cancer cells.**

**(A)** Gα$_{i3}$ expression in PANC-1 and HCT116 cells, evaluated by IF. Representative images (scale bar: 100 μm) are on the left, and the fluorescence intensity bar graph is on the right. Data were obtained from the analysis of 20 cells per condition. Data are mean values ± s.e.m. (n = 3). a.u., arbitrary units. **(B)** Girdin expression in PANC-1 and

in HEK–hERG1 cells was operant in cancer cells. $G\alpha_{i3}$ (Fig 7A) and girdin (Fig 7B) were expressed in our cancers cells seeded on FN ($T_{90}$), at levels comparable to those detected in HEK–hERG1 cells (Fig 3), and both co-localized with $\beta 1$ integrins. Co-localization peaked at $T_{90}$ and decreased thereafter (Fig 7C and D). Both in PANC-1 and HCT116 cells, PTX inhibited $G\alpha_{i3}/\beta 1$ integrin co-localization (Fig 7E, left panel), the hERG1 translocation to the plasma membrane evidenced by FC (Fig 7E, middle panel), and the hERG1/$\beta 1$ integrin complex formation evidenced by co-IP (Fig 7E, right panels). Fig 7E shows data from densitometric analysis, all the original data are in Fig S7. Similar effects were produced by silencing girdin (Fig S7), which decreased the hERG1 translocation to the plasma membrane evidenced by FC (Fig 7F) and the hERG1/$\beta 1$ integrin complex assembly evidenced by IF staining with fluorescent scDb–hERG1–$\beta 1$ (Fig 7G).

To clarify whether the conductive role of hERG1 is implicated in the signaling mechanisms activated by the hERG1/$\beta 1$ integrin complex formation, we compared the effects of treating live cells with either the hERG1 channel blocker E4031 or scDb–hERG1–$\beta 1$. The latter impairs formation of the hERG1/$\beta 1$ integrin complex (Fig 5F and Duranti et al, 2021b), without blocking the current (Fig 8A). Both co-IP and IF experiments in PANC-1 and HCT116 cells seeded on FN for 90 min showed that scDb–hERG1–$\beta 1$ was even more effective than E4031 in impairing hERG1/$\beta 1$ complex formation (Figs 8B and C and S8A). Consistently, scDb–hERG1–$\beta 1$ produced significantly higher effects than E4031 in impairing hERG1 translocation to the plasma membrane (Fig 8D), and $\beta 1$ integrin co-localization with G$\alpha$i3 (Fig 8E) and girdin (Fig 8F) in both cancer cell types (Fig S8B–D). These results suggest that integrin-dependent signaling is mainly impaired by the disruption of the hERG1/$\beta 1$ integrin complex irrespectively of hERG1 current block. Our interpretation of these results will be discussed later.

### A biochemical network model of the dynamics of hERG1 and $\beta 1$ integrin interaction

From the collected experimental observations, we elaborated a comprehensive kinetic model that should contribute to elucidate the inner workings of the signaling pathway that we found to regulate hERG1 expression cycle (Fig 9 and Supplemental Data 1). An essential prerequisite for the comparison of theory with experimental observations was the quantification of selected kinetic experimental readouts in terms of molecules/cell (Table 2). In our model (Fig 9A), *herg1* mRNA is synthesized and degraded. The channel is synthetized in the ER, and then transferred to the Golgi apparatus, where it undergoes maturation by glycosylation, and is eventually translocated to the plasma membrane. On the cell surface, the free channel can transition between open and closed states. The closed free (i.e., not complexed with $\beta 1$ integrin) channel can be internalized and directed to the (late) endosomal system, be directly degraded, or bound to $\beta 1$ integrins (active or inactive). The channel is assumed to complex with integrins only in the closed state and not to be trafficked within the cytoplasm other than through the afore-described maturation/translocation pathway. These assumptions reflect our experimental results (Figs 1–7 of the present study and Becchetti et al, 2017) and are in line with general aspects of receptor trafficking (Lauffenburger & Linderman, 1993). In particular, hERG1 molecules in the endosomes can either go back to the membrane or be degraded. Integrins can associate and dissociate with/from FN to become active/inactive. Active integrins (bound or not to the channel) enhance catalytically the synthesis of the channel mRNA. We incorporated in our model a generic balanced inactivation–like module (Fig 9B): integrin activation triggers two contrary responses: a rapidly increasing excitation and a slowly increasing inhibition (Levine et al, 2006). In particular, we introduced one activator (A) and one inhibitor (I), meant to represent coarse-grained versions of only partially identified biochemical subnetworks, which we assumed to be both catalytically controlled by the formation of hERG1/$\beta 1$ integrin complex. The inhibitor inactivates the activator, whereas the activator enhances several of the aforementioned reactions, namely, translation, import to and export from the Golgi, and also inhibits channel internalization/degradation. For the sake of simplicity, the model assumes that all components are homogeneously distributed in a volume corresponding to that of a single cell (i.e., without compartmentation), thus taking the form of a set of coupled differential equations for time-varying copy numbers of molecular species/states (see Appendix). Correspondingly, second-order reaction rates have units of (molecules/cell)$^{-1}$ s$^{-1}$. It should be noted that the association and dissociation rates of FN and integrins in our model may not reflect the true rates, as the cell adhesion dynamics may be a cooperative process (Pouwels et al, 2012).

Overall, this model is built around the concept of a signal that simultaneously triggers a signaling response and its inhibition. It is

---

HCT116 cells, evaluated by IF. Representative images (scale bars: 100 $\mu m$) are on the left, the fluorescence intensity bar graph is on the right. Data were obtained from the analysis of 20 cells per condition. Data are mean values ± s.e.m. (*n* = 3), a.u. **(C)** Time course of $\beta 1$ integrin and G$\alpha_{i3}$ co-localization in PANC-1 and HCT116 cells, evaluated by IF. The corresponding Manders' Overlap coefficient quantification is in the graph on the right. Data were obtained from the analysis of 20 cells per condition. Data are mean values ± s.e.m. (*n* = 3). **(D)** Time course of $\beta 1$ integrin and girdin co-localization in PANC-1 and HCT116 cells, evaluated by IF. Representative images are on the right (scale bars: 100 $\mu m$). The corresponding Manders' Overlap coefficient quantification is in the graph on the right. Data were obtained from the analysis of 20 cells per condition. Data are mean values ± s.e.m. (*n* = 3). **(E)** Effects of PTX on PANC-1 and HCT116 cells. Left panel: IF co-localization of $\beta 1$ integrin and G$\alpha_{i3}$ in PANC-1 and HCT116 untreated (CTR) or treated with PTX (PTX). Data were obtained from the analysis of 20 cells per condition. Data are mean values ± s.e.m. (*n* = 3). Middle panel: hERG1 mean fluorescence intensity in PANC-1 and HCT116 untreated (CTR) or treated with PTX (PTX). Data are reported as fold increase in a. u. between 0 and 3.0. Data are mean values ± s.e.m. (*n* = 3). a.u. Right panel: hERG1/$\beta 1$ integrin complex formation in PANC-1 and HCT116 cancer cells. Data are reported as fold increase in a. u. between 0 and 3.0. Data are mean values ± s.e.m. (*n* = 3), a.u. **(F)** Effects of girdin silencing on hERG1 translocation on the plasma membrane assessed by FC in PANC-1 and HCT116 cells seeded on FN for 90 min. Representative plots are in the top panel, whereas the hERG1 mean fluorescence intensity of the area under the curve is reported in the bottom panel. Data have been normalized on the zero value and reported as fold increase a.u. on a scale ranging between 0 and 3.0. Data are mean values ± s.e.m. (*n* = 3). **(G)** Effects of girding silencing on hERG1/$\beta 1$ complex formation in PANC-1 and HCT116 cells seeded on FN for 90 min. Representative IF images (top panel) (scale bars: 100 $\mu m$) and IF densitometric analysis (bottom panel) are reported. Data were obtained from the analysis of 20 cells per condition for IF analysis. Data are mean values ± s.e.m. (*n* = 3), a.u. Fig 7 shows representative data, the whole set of data is in Fig S7.

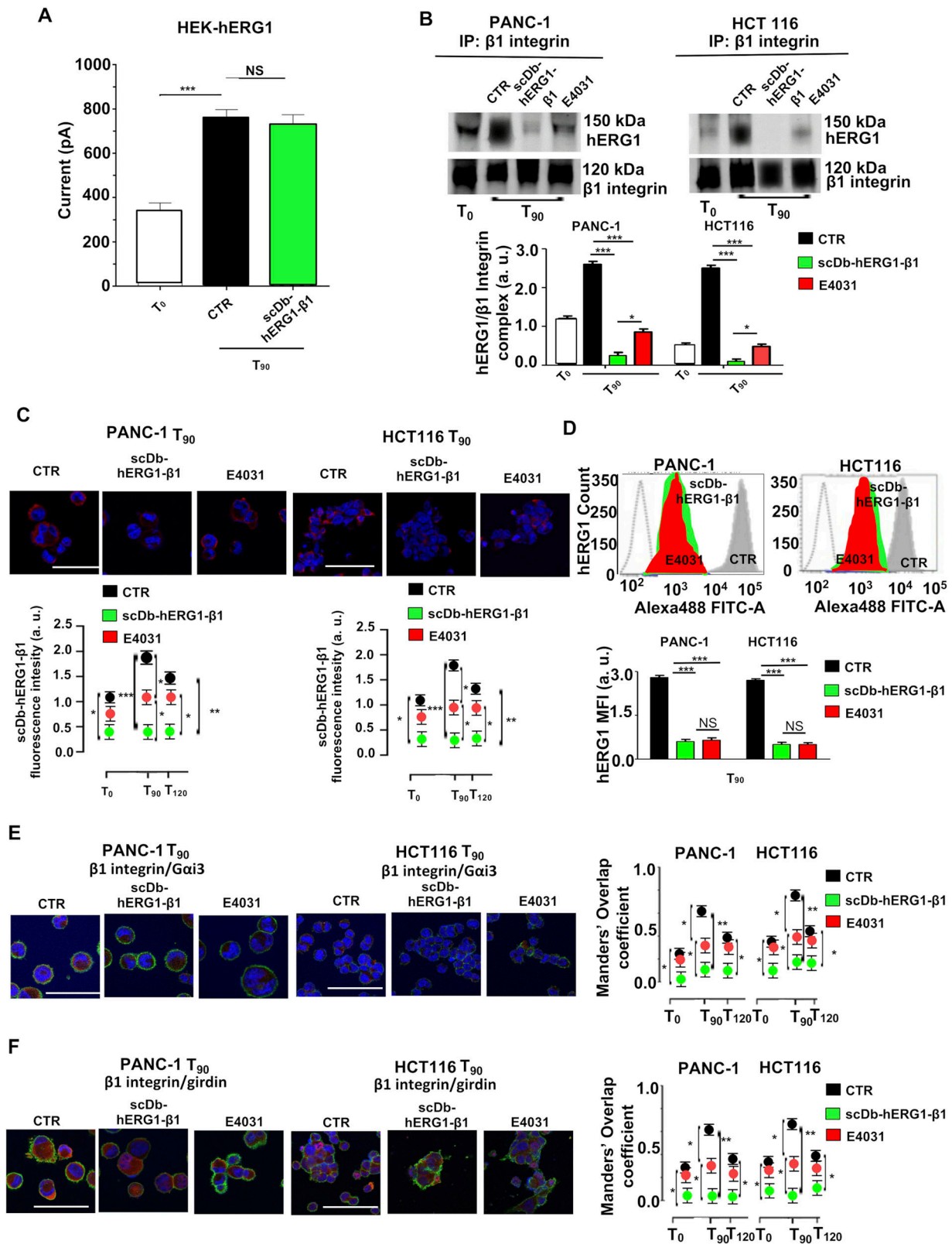

**Figure 8. Effects of scDb–hERG1–β1 and E4031 on hERG1/β1 complex formation and downstream signaling in PANC-1 and HCT116 cells.**
**(A)** $I_{hERG1}$ in HEK–hERG1 cells stimulated for 90 min on FN and treated for 90 min with scDb–hERG1–β1 (20 μg/ml). **(B, C, D, E, F)** Effects of treatment with scDb–hERG1–β1 (20 μg/ml) or E4031 (40 μM) of PANC-1 and HCT116 live cells seeded on FN for different times. **(B)** Co-immunoprecipitation of hERG1 with β1 integrins at $T_{90}$. Representative blots are in the top panels, the densitometric data are in the bottom panels. Data have been normalized on the zero value and reported as fold increase arbitrary units

thus designed to lead to an activation that dominates only in the short term following the signal onset. This is the behavior displayed by the experimental system, which shifts nonlinearly between two different stationary states, respectively, in the absence/presence of FN or any other activating mechanism (see Figs 1 and 2). Because the two stationary states are different, there must be some part of the signal that triggers another kind of non-inhibited response. This seems to be the case of the channel's mRNA, whose amount progressively increases as the system progresses from the first to the second stationary state (see Fig 2A) and seemingly without displaying a peaked behavior, differently from the channel proteins themselves (Fig 2B–D). According to this observation, we assume that the mRNA synthesis boost is directly stimulated by integrin activation, rather than by the coarse-grained activator species. In our model, the activator, generated upon integrin activation, boosts translation of the *herg1* mRNA, trafficking of hERG1 molecules from the ER to the Golgi and translocation to the plasma membrane. Moreover, guided by the observation that the translocation peak corresponds to a negative peak in hERG1 internalization (from the co-localization with Rab5), we introduce an unknown internalization module that also responds to the integrin activation signal and whose inhibition is catalyzed by the activator (see Fig 9A). The mathematical details are reported in the Supplemental Data 1.

Fig 10 shows that our model reproduces the available experimental data with good accuracy, in HEK–hERG1 cells and in PANC1 and HCT116 cancer cell lines. Our model overall confirms that a signal triggered by integrin activation that boosts *directly* hERG1 transcription and *indirectly and transiently* (through the activator) translation, trafficking and inhibition of internalization can explain the complex pattern of experimental observations.

### The functional consequences of the hERG1/$\beta$1 integrin interaction in cancer cell signaling and migration

Finally, we studied how the interaction of $\beta$1 integrins with hERG1 affected cell signaling in our cell models. Based on the results in Fig 3G and the known link between girdin and Akt (Anai et al, 2005; Enomoto et al, 2005; Ghosh et al, 2008; Jiang et al, 2008; Kitamura et al, 2008), we first focused on the PI3K/Akt pathway. In HEK–hERG1 and cancer cell lines, the pAkt amount increased from $T_0$ to $T_{90}$ after cell adhesion to FN (Figs 11A and S9, white bars), with a kinetics similar to that displayed by the hERG1/$\beta$1 integrin complex formation (see Figs 4 and 6). No concomitant variations of ERK phosphorylation (pERK) were observed (Fig 11A, black bars). The pAkt peak at $T_{90}$ was impaired by PTX both in HEK–hERG1 and in

cancer cell lines (Fig 11B). No effects of PTX were observed on pERK (Fig 11B).

Next, we tested whether the formation of the hERG1/$\beta$1 integrin complex modified f-actin organization. To this purpose, cells were stained with rhodamine-conjugated phalloidin, and both the length of stress fibers (Pier et al, 2014; Manoli et al, 2019) and the cortical f-actin density (Leyme et al, 2015) were determined. In HEK–hERG1 cells seeded on FN, stress fibers length slightly decreased from $T_0$ to $T_{90}$. Conversely, the average cortical f-actin density increased from $T_0$ to $T_{90}$ (Fig 11C). These effects were partially reversed by PTX, which slightly increased stress fibers length and decreased cortical f-actin density (Fig 11D). Inhibiting the FN-triggered signaling pathway described so far with PTX, E4031, or scDb-hERG1-$\beta$1 in PANC-1 and HCT 116 cells modified f-actin organization, increasing stress fibers length and decreasing cortical f-actin density (Fig 11E and F). These effects were stronger in cells treated with the scDb–hERG1–$\beta$1, which dissociates the hERG1/$\beta$1 integrin complex, but does not block the hERG1 currents (Fig 8B). Overall, these data indicate that, once complexed with hERG1, $\beta$1 integrin works as a signaling hub, mainly affecting the PI3K/Akt pathway and the organization of f-actin.

Finally, we analyzed whether the above change of f-actin organization could alter cancer cell pro-migratory behavior. To this purpose, both the lateral motility and the FN-induced haptotaxis were measured in PANC-1 and HCT116 cells (i) where girdin was silenced, (ii) treated with PTX, (iii) treated with the scDb–hERG1–$\beta$1, or (iv) treated with E4031. All these treatments, which inhibited the FN-triggered signaling pathway described so far, significantly reduced lateral motility (Fig 11G) and haptotaxis (Fig 11H). Once again, these effects were stronger in cells treated with the scDb–hERG1–$\beta$1, compared with E4031.

## Discussion

By combining experimental data and quantitative modeling, we defined a novel signaling pathway by which $\beta$1 integrins and hERG1 interact to regulate f-actin organization and promote cell migration. Integrin engagement by the ECM protein FN stimulates a girdin-dependent activation of trimeric G$\alpha$i3 proteins, and thus of PI3K/Akt, which promotes hERG1 translocation to the plasma membrane and assembly of the hERG1/$\beta$1 integrin complex. By sequestering closed hERG1 channels, $\beta$1 integrins make hERG1 avoid the Rab5-mediated endocytic pathway and the ensuing degradation. Moreover, the hERG1/$\beta$1 integrin complex stimulates cell motility/

---

(a.u.) on a scale ranging between 0 and 3.0. Data are mean values ± s.e.m. ($n$ = 3). **(C)** hERG1/$\beta$1 complex formation evaluated by staining with fluorescent scDb–hERG1–$\beta$1. Representative IF images at $T_{90}$ (top panel) (scale bars: 100 $\mu$m) and IF densitometric analysis at different time points (bottom panel) are reported. Data have been normalized on the zero value and reported as fold increase a.u. on a scale ranging between 0 and 2.5. **(D)** hERG1 plasma membrane translocation at $T_{90}$. Representative FC plots are in the top panel, the hERG1 MFI is in the bottom panel. Data have been normalized on the zero value and reported as fold increase a.u. on a scale ranging between 0 and 3.0. **(E)** $\beta$1 integrin and G$\alpha_{i3}$ co-localization evaluated by IF. Representative images at $T_{90}$ are on the left (scale bars: 100 $\mu$m), the corresponding Manders' Overlap coefficient quantification at different time points are in the graph on the right. **(F)** $\beta$1 integrin and girdin co-localization evaluated by IF. Representative images at $T_{90}$ are on the left (scale bars: 100 $\mu$m), the corresponding Manders' Overlap coefficient quantification at different time points are in the graph on the right. All the IF data were obtained from the analysis of 20 cells per condition and are reported as fold increase in a.u. between 0 and 3.0. Data are mean values ± s.e.m. from three different experiments ($n$ = 3). NS, not significantly different, \*$P$ < 0.05, \*\*$P$ < 0.01, and \*\*\*$P$ < 0.001. Fig 8 shows representative data, the whole set of data is in Fig S8. Source data are available for this figure.

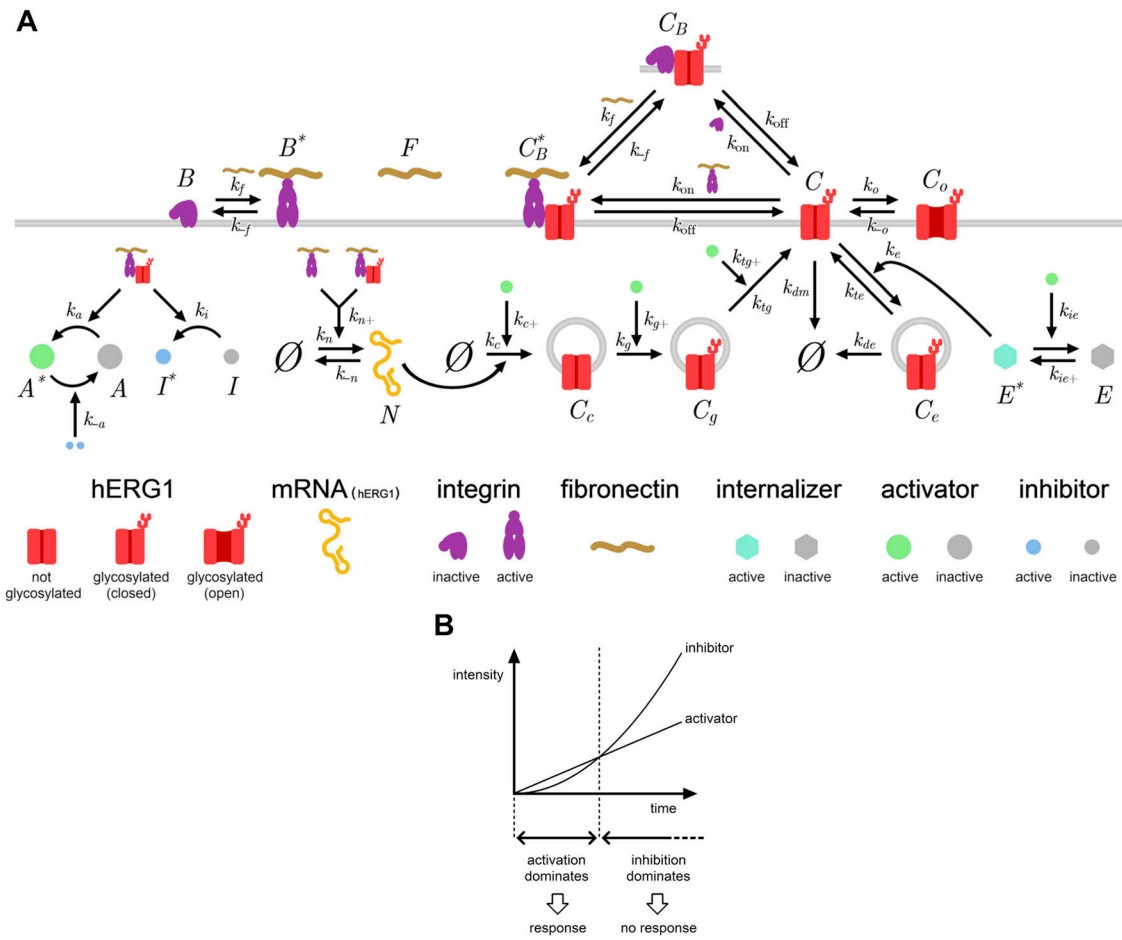

**Figure 9. Coarse-grained biochemical network model.**
**(A)** Cartoon of the coarse-grained network model developed to describe the enhanced transcription, translation, trafficking, complex formation, and degradation of hERG1 in transfected HEK and cancer cells. **(B)** General idea behind the balanced inactivation–like module that underpins the novel signaling pathway unveiled by our experiments. The parameters below the grey band have been kept fixed during the fits, as their values did not impact significantly the kinetics of propagation of the signal triggered by integrin–hERG1 complex formation.

haptotaxis through an f-actin–dependent mechanism. The experimental parameters were used to develop a mathematical model, based on a generic balanced inactivation–like module. Integrin engagement triggers two contrary responses with different kinetics: a rapid stimulation of hERG1 translocation, which determines an increase in $I_{hERG1}$, $V_{rest}$ hyperpolarization, and in turn triggers hERG1/$\beta$1 complex formation. This phase is then followed by a slowly developing inhibition that restores the resting state within 300 min. Overall, the present study explains (i) the mechanism underlying the aberrant expression of hERG1 in cancer, (ii) how integrin receptors stimulate $I_{hERG1}$, and (iii) how hERG1 and $\beta$1 integrins interact to produce a slow cycle of expression on the plasma membrane, which matches the protrusion/retraction cycle in cell migration.

### Integrin-dependent hERG1 overexpression and $I_{hERG1}$ modulation in cancer

We show that $\beta$1 integrin activation increases $I_{hERG1}$ amplitude by stimulating hERG1 expression and translocation to the plasma membrane, without altering the hERG1 biophysical properties. This may explain the frequent hERG1 overexpression in cancer which often occurs without any evident genetic or epigenetic alterations (Crociani et al, 2014; Arcangeli et al, 2023). Based on our data, the tumor microenvironment, through the engagement of cellular integrins by specific ECM proteins, would up-regulate hERG1 expression in the plasma membrane. This could occur both in solid cancers (Arcangeli et al, 2023) and in leukemias (Pillozzi et al, 2011).

hERG1 translocation is controlled by the trimeric G protein G$\alpha$i3, through the PI3K/Akt pathway. The implication of Akt is consistent with previous reports (Zhang et al, 2003; Sangoi et al, 2017; Wang et al, 2019), whereas the possible involvement of the chaperon SigmaR (Crottès, et al, 2011) in our model remains to be determined. G$\alpha$i3 activation is driven by the non-receptor GEF girdin (Garcia-Marcos et al, 2015). Notably, the kinetics of G protein activation by girdin is considerably slower than the classical GPCR–dependent mechanism (Aznar et al, 2016). This is in line with the slow kinetics of hERG1 cycling, which is sustained by the long lifetime of the hERG1/$\beta$1 integrin complex. Moreover, the latter preferentially recruits closed channels, thus slowing down hERG1 degradation. Closed

**Table 2. The best-fit values of the floating parameters of the model.**

| Parameter | HEK–hERG1 | PANC-1 | HCT116 | Units |
|---|---|---|---|---|
| $k_c$ | 0.402 | 1.43 | 1.5 | $s^{-1}$ |
| $k_g$ | 0.0524 | 0.139 | 0.205 | $s^{-1}$ |
| $k_{tg}$ | 0.00304 | 0.00206 | 0.00154 | $s^{-1}$ |
| $k_e$ | $3.55 \times 10^{-5}$ | 0.000227 | 0.000352 | $(\#molecules/cell)^{-1}s^{-1}$ |
| $k_{te}$ | 0.0608 | 0.0218 | 0.0292 | $s^{-1}$ |
| $k_{de}$ | 0.105 | 0.18 | 0.253 | $s^{-1}$ |
| $k_{off}$ | 107 | 346 | 234 | $s^{-1}$ |
| $k_{on}$ | $4.24 \times 10^{-5}$ | 0.000113 | $8.82 \times 10^{-5}$ | $(\#molecules/cell)^{-1}s^{-1}$ |
| $k_{-f}$ | $8.05 \times 10^{-6}$ | $8.37 \times 10^{-5}$ | $8.79 \times 10^{-5}$ | $s^{-1}$ |
| $k_a$ | $4.91 \times 10^{-8}$ | $1.18 \times 10^{-7}$ | $1.92 \times 10^{-7}$ | $(\#molecules/cell)^{-1}s^{-1}$ |
| $k_{a-}$ | 0.000258 | 0.000873 | 0.000486 | $(\#molecules/cell)^{-2}s^{-1}$ |
| $k_i$ | $1.23 \times 10^{-9}$ | $1.17 \times 10^{-9}$ | $1.73 \times 10^{-9}$ | $(\#molecules/cell)^{-1}s^{-1}$ |
| $k_{ie}$ | 65.2 | 79.2 | 71.8 | $(\#molecules/cell)^{-1}s^{-1}$ |
| $k_n$ | 163 | 100 | 86.4 | $(\#molecules/cell)^{-1}s^{-1}$ |
| $k_{n+}$ | 0.000776 | 0.000898 | 0.00109 | $s^{-1}$ |
| $k_{-n}$ | 0.0134 | 0.01 | 0.0116 | $s^{-1}$ |
| $k_{c+}$ | 0.00804 | 0.0286 | 0.03 | $(\#molecules/cell)^{-1}s^{-1}$ |
| $k_{g+}$ | 0.000698 | 0.00185 | 0.00274 | $(\#molecules/cell)^{-1}s^{-1}$ |
| $k_{tg+}$ | 0.000108 | $7.32 \times 10^{-5}$ | $5.46 \times 10^{-5}$ | $(\#molecules/cell)^{-1}s^{-1}$ |
| $k_{dm}$ | 0.105 | 0.18 | 0.253 | $s^{-1}$ |
| $k_f^0$ | $1.75 \times 10^{-5}$ | $6.29 \times 10^{-6}$ | $8.42 \times 10^{-6}$ | $(\#molecules/cell)^{-1}s^{-1}$ |
| $k_{ie+}$ | 0.132 | 0.225 | 0.317 | $s^{-1}$ |
| $k_{o-}$ | 8 | 8 | 8 | $s^{-1}$ |
| $k_o$ | 0.15 | 0.15 | 0.15 | $s^{-1}$ |
| $B_0$ | $1.709 \times 10^7$ | $1.709 \times 10^7$ | $1.709 \times 10^7$ | $\#molecules$ |
| $A_0$ | $1.5 \times 10^3$ | $1.5 \times 10^3$ | $1.5 \times 10^3$ | $\#molecules$ |
| $I_0$ | $1.5 \times 10^3$ | $1.5 \times 10^3$ | $1.5 \times 10^3$ | $\#molecules$ |
| $E_0$ | $10^3$ | $10^3$ | $10^3$ | $\#molecules$ |
| $F_0$ | $2.4 \times 10^5$ | $2.4 \times 10^5$ | $2.4 \times 10^5$ | $\#molecules$ |

The last seven bottom lines report parameters that have been kept at a fixed value during the fits.

channels bound to the complex are less sensitive to RAB5-mediated endocytosis (Figs 4F and 5I), the main route for degradation of the non-complexed hERG1 (Foo et al, 2016). Finally, the results obtained with the hERG1 channel blocker E4031 and scDb–hERG1–$\beta$1 (Fig 8) lead us to conclude that the role of hERG1 in the signaling pathway downstream to integrin activation in cancer cells is mainly related to the promotion of hERG1/$\beta$1 integrin complex formation, without a major contribution of K$^+$ current. This interpretation is suggested by the following observations. First, the complex formation is hampered by scDb–hERG1–$\beta$1, which disrupts the hERG1/$\beta$1 integrin interaction (Duranti et al, 2021b) without blocking the current (Fig 8A). Second, E4031 also impairs the complex formation but blocks the channel in the open state, which has a lower affinity for $\beta$1 integrin (Becchetti et al, 2017). Such reasoning is consistent with the results obtained with the K525C hERG1 mutant that preferentially resides in the open state

(Becchetti et al, 2017 and Fig 5). Nonetheless, a rigorous definition of the role of hERG1 current in ECM-activated signals will require direct structural studies about the scDb–hERG1–$\beta$1 interaction with the complex and hERG1 interaction with $\beta$1 integrin in different conformational states.

## Insights from the mathematical modeling of experimental data

We built up a mathematical model, by applying ordinary differential equations to selected kinetic experimental readouts which were first quantified in terms of molecules/cell. The results of our global fit lead to a good agreement between the best-fit model and the available kinetic data (Fig 10). Because of the limited information available on many microscopic reaction rates, we have evaluated the sensitivity of the results to variations of the best-fit values of the floating parameters. The shaded regions in Fig 10 correspond to

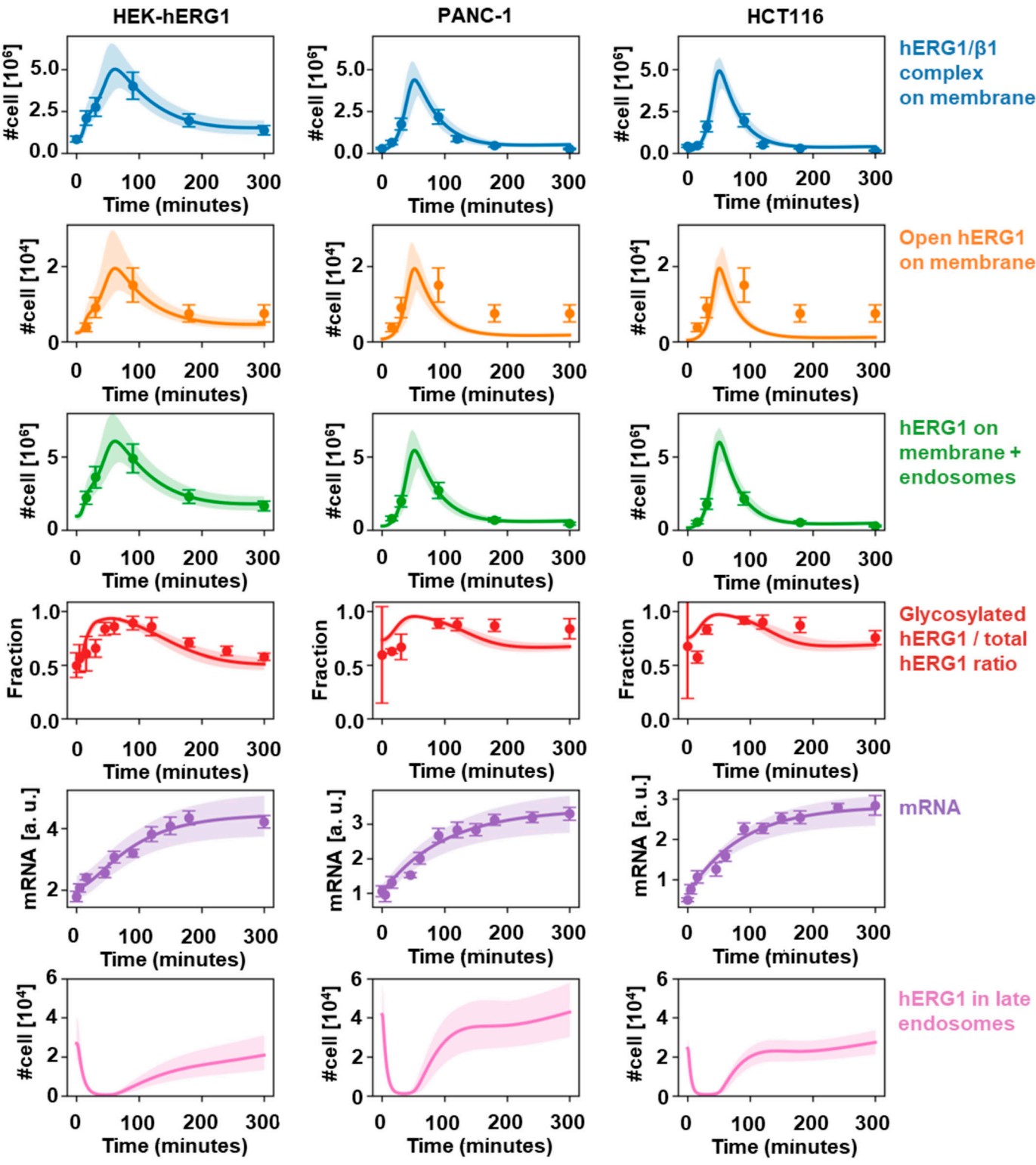

**Figure 10. Fitting of experimental data.**
Expression, translocation, and complex formation of hERG-1 upon adhesion of HEK cells on fibronectin. Comparison of the experimental data (symbols) with the best-fit solution of the rate equations (see supplementary PDF document, solid lines). In particular, the number of hERG1/β1 integrin complexes (from co-IP experiments), the fraction of glycosylated versus total hERG1 channels (from WB data), the number of open channels (from I$_{hERG1}$ data); the number of RNA molecules (from RQ-PCR data), the number of hERG1 channels localized on the plasma membrane and microdomains close to the membrane/late endosomes (from IF data) (see supplementary pdf document). The shaded areas mark the observed variability regions corresponding to 100 different independent sets of the floating parameters obtained by randomly perturbing each of them within ±10% of their best-fit value. The three columns refer to the three cell lines analyzed in this work. The last row of plots depicts the predicted

the results obtained by considering random combinations of the free parameters chosen to lie within ± 10% of the corresponding best-fit results. It can be seen that such random fluctuations approximately reproduce the same order of magnitude of variability displayed by the experimental uncertainties as fixed by the error bars. A full stochastic analysis, which is outside the scope of the present work, will likely describe more accurately the dynamics of tumor cells, where the numerosity of the relevant molecular species is lower.

Our kinetic model sheds light into the dynamics of hERG1 and hERG1/$\beta$1 integrin complex trafficking. We included a catalytically augmented internalization signal to reproduce the internalization peak that runs quasi-simultaneously to the peak displayed by the hERG1–integrin complex. In particular, we posited the existence of a catalytic module that governs the internalization of free hERG1 molecules (CG species $E$) but, that is, *inhibited* by the signal provided by the species A*, that is, activated by the hERG1–integrin complex. When the short pulse that describes the catalytic-balanced activation module (A,A*-I,I*) starts relaxing, the catalytically governed internalization of free hERG1 molecules regains strength and late endosomes appear to start filling up again (see last row of panels in Fig 10).

In addition, our theoretical model provides insights into the differences that characterize the integrin-dependent augmented hERG1 expression and translocation in normal and cancer cells. First, the levels of *herg1* mRNA and hERG1 protein evolve on different time scales in the experiments, mRNA being produced at a much slower pace with respect to protein degradation through the RAB5-mediated pathway. Such different dynamics are largely due to the nonlinear dynamics of the activator-inhibitor trigger over time in the presence of active integrin. In the short term, when the activator dominates, there is a boost in translation and trafficking, which disappears in the long term when the inhibitor dominates. In summary, in addition to active degradation, there also appears to be a nonlinear regulation mechanism of synthesis and trafficking, which helps set up the hERG1 peak at T$_{90}$ min. Second, although the mRNA copy number is lower in cancer cells compared with the over-expressing HEK–hERG1, the number of complexes on the plasma membrane at the peak is predicted to be of the same order of magnitude in either cell type. Although translation is an inherently nonlinear process, this observation appears non-trivial. In particular, our fits find that the activation rate $k_a$ that transduces the signal coming from the hERG1–integrin complexes is about 10 times larger in cancer cells than in HEK–hERG1. This suggests that, in cancer cells, this signaling module is swifter in transducing the signal. This could compensate the slower translation due to the lower *hERG1* mRNA.

The dynamics of hERG1 translocation is also different in HEK–hERG1 and cancer cells. More precisely, the relaxation to the post-integrin activation stationary state is faster in cancer cells, not only because the complexes dissociate more readily ($k_{off}$ is two to three times faster in cancer cells; Fig 9A, Table 2 and Supplemental Data 1) but also because of a different kinetics of hERG1 internalization. In cancer cells, the endocytosis rate $k_e$ is higher, although the backward translocation associated with the late endosomes in the proteasome pipeline ($k_{te}$) is slower (Fig 9A, Table 2 and Supplemental Data 1). These two effects combine to make the hERG1 localization peak on the plasma membrane to relax more quickly in cancer cells. This difference could be related to the cancer phenotype or be a "trivial" consequence of hERG1 overexpression in HEK–hERG1.

Finally, a relevant feature of our model is that integrin activation triggers two contrary responses: a relatively rapidly increasing stimulation and a slowly increasing inhibition (Levine et al, 2006) (Fig 9B). Both the activator and the inhibitor are catalytically controlled by the formation of hERG1/$\beta$1 integrin complex. The activator dominates following the signal onset, enhancing several of the aforementioned reactions (e.g., translation, import to and export from the Golgi) and inhibits channel internalization. The inhibitor inactivates the activator. The activator (A) and the inhibitor (I) are meant to represent coarse-grained versions of thus far partially identified biochemical subnetworks. Based on experimental data, the former is likely to be constituted by the girdin–G$\alpha$i3–G$\beta\gamma$–PI3K–Akt pathway, whereas the molecular correlate(s) of the inhibitor have not been identified so far, although we could hypothesize the involvement of hERG1/$\beta$1 complex internalization by a slower endocytic pathway.

### Conclusion: hERG1/$\beta$1 integrin interaction in cancer cell migration

Our findings delineate a novel mechanism, based on the hERG1–$\beta$1 integrin interaction, whose dynamics shows biphasic behavior: (1) integrin engagement stimulates hERG1 channel translocation to the plasma membrane, which increases I$_{hERG1}$ amplitude and causes V$_{rest}$ hyperpolarization; (2) hERG1 assembles with the integrin and remains in the plasma membrane (i.e., is not degraded) in the closed conformation, which progressively restores the initial V$_{rest}$. The hERG1/$\beta$1 integrin complex and the signaling mechanism that sustains its formation (based on girdin-dependent G$\alpha$i3 activation) have a striking impact on cell migration. This can be traced back to an effect on f-actin, both on stress fibers length and cortical f-actin organization. The latter is known to affect the contractile machinery in normal (Alibert et al, 2017; Warmt et al, 2021) and cancer cells, which usually show a more disorganized cytoskeleton and lower cortical tension (Efremov et al, 2014, Svitkina, 2020; Hosseini et al, 2020). Although, the exact mechanisms regulating the biomechanics of cortical f-actin in normal and cancer cells have not been fully clarified yet, we provide evidence that they include the hERG1/$\beta$1 integrin complex.

time course of the hERG1 population in late endosomes, showing qualitative agreement with the results of Rab5 co-localization experiments (see Fig 4D). The symbols relative to the number of open hERG1 on membrane refer to experimental data obtained in HEK–hERG1 cells. The experimental points in the second row are those from HEK cells for all the three cell lines. In the case of PANC-1 and HCT116 cells, similar data were not available and were not included in the cost function used to fit simultaneously the experimental readouts but are nonetheless shown here for comparison. Thus, for those cell lines, the solid lines should be interpreted as the time course of the number of open hERG1 channels on the tumor cells' membranes predicted by our model.

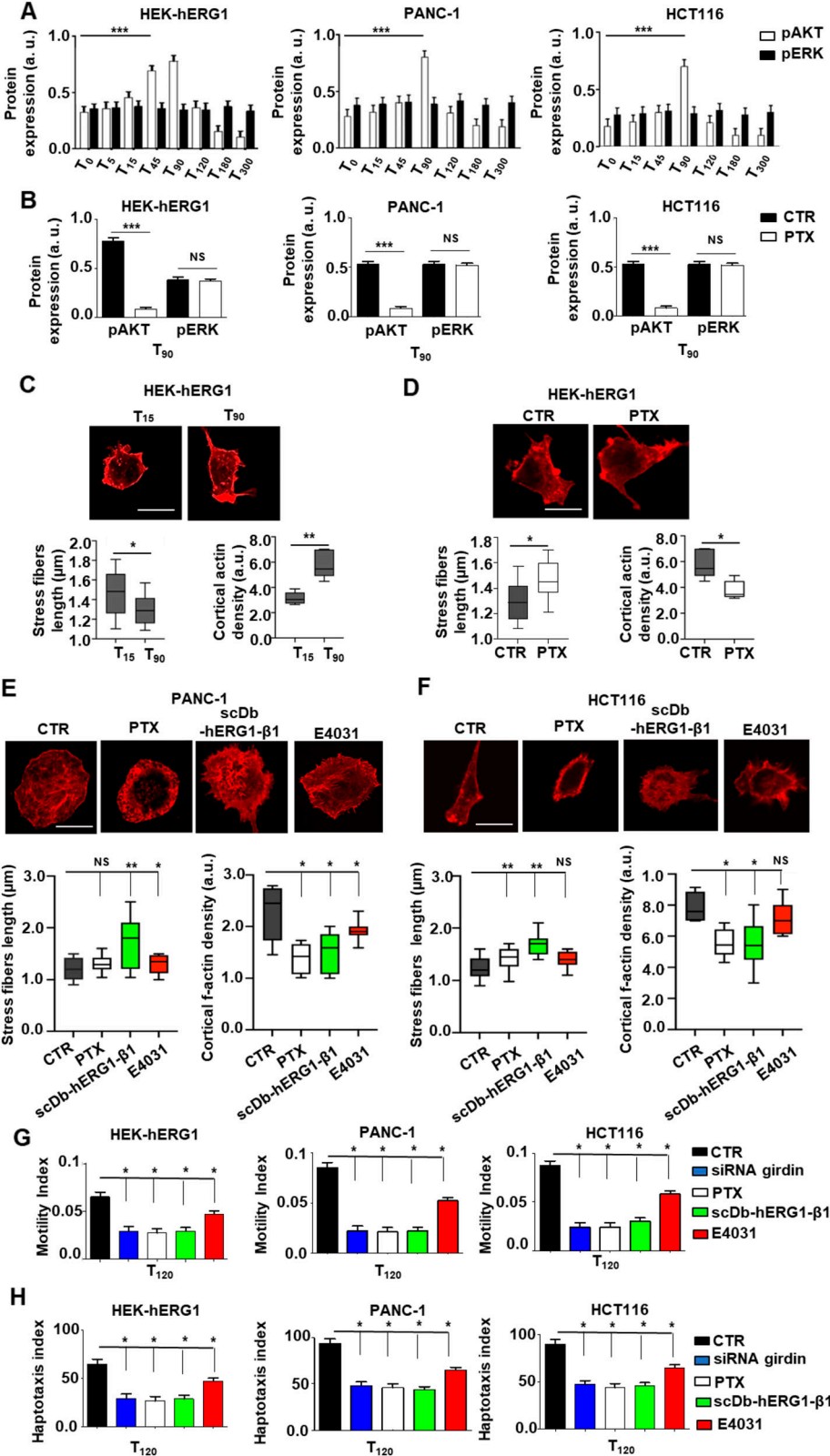

**Figure 11. Functional relevance of hERG1/β1 integrin complex on Akt signaling, f-actin organization, and cell motility and invasiveness in cancer cells.**

**(A)** Time course of Akt (white bars) and Erk 1/2 (black bars) phosphorylation in HEK–hERG1, PANC-1, and HCT116 cells seeded on FN. The bar graph shows the densitometric analysis of p-Akt $^{Thr308}$ and p-ERK1/2$^{Thr202/Tyr204}$ relative to the not phosphorylated forms of the two proteins. Data are mean values ± s.e.m. ($n$ = 3). a.u., arbitrary units. The slopes of the lines fitting experimental values between $T_0$ and $T_{90}$ for pAkt versus pERK in the different cell lines were: 0.37 versus −0.015 (HEK–hERG1); 0.52 versus 0.01 (PANC-1); 0.52 versus 0.005 (HCT116). **(B)** Effects of PTX treatment on p-Akt $^{Thr308}$/AKT and p-ERK1/2$^{Thr202/Tyr204}$/ERK 1/2 proteins in HEK–hERG1, PANC-1, and HCT116 cells seeded on FN at $T_{90}$. Data are mean values ± s.e.m. ($n$ = 3). a.u., arbitrary units. **(C)** Phalloidin staining of HEK–hERG1 at $T_{15}$ and $T_{90}$ after cell seeding on FN. Representative images are on the upper panels, the corresponding graphs of actin stress fiber length (left panel) and cortical actin density (right panel) are on the bottom. **(D)** Phalloidin staining of HEK–hERG1 after cell seeding on FN at $T_{90}$ (CTR) and treated with PTX. Representative images are on the upper panels, the corresponding graphs of actin stress fiber length (left panel) and cortical actin density (right panel) are on the bottom. **(E)** Phalloidin staining of PANC-1 cells after seeding on FN at $T_{90}$ (CTR), pre-treated overnight with 100 mg/ml of PTX (PTX), and treated with scDb–hERG1–β1 (20 μg/ml) and E4031 (40 μM). Representative images are on the upper panels, the corresponding graphs of actin stress fiber length (left panels) and cortical actin density (right panels) are shown in the bottom. **(F)** Phalloidin staining of HCT116 cells after seeding on FN at $T_{90}$ (CTR), pre-treated overnight with 100 ng/ml of PTX and treated with scDb–hERG1–β1 (20 μg/ml) and E4031 (40 μM). Representative images are on the upper panels, the corresponding graphs of actin stress fiber length (left panels) and cortical actin density (right panels) are shown in the bottom. In all the graphs, boxes include central 50% of data points, and the horizontal lines denote minimum value, median, and maximum value. At least a total of 20 cells per condition from three independent experiments were analyzed, and all $P$-values were determined by a Mann–Whitney test (significant level set to $P < 0.05$), or for data deviating from normality by a Kolmogorov–Smirnov test. Scale bars: 10 μm. **(G, H)** Motility index and (H) Haptotaxis index of HEK–hERG1, PANC-1, and HCT116 seeded on FN (CTR) for 90 min after girdin silencing, pre-treatment with PTX, treatment with scDb–hERG1–β1 or with E4031. Motility index (MI) was assessed using the following formula: MI = 1 − Wt/W0, where Wt is the width of the wounds. The Haptotaxis index refers to the mean number of migrating cells counted in five optical fields. Data are presented as mean values ± s.e.m. ($n$ = 3). NS, not significantly different, *$P < 0.05$, **$P < 0.01$, and ***$P < 0.001$. All the original images and WBs relative to Fig 11 are shown in Fig S9.

Finally, the signaling mechanisms regulating hERG1 cycling in the plasma membrane appears to be gauged to match the time-course of the protrusion–retraction cycles of cell migration stages, whose kinetics is in the order of tens of minutes (Seetharaman & Etienne-Manneville, 2020; Adebowale et al, 2021). In the context of cancer growth in vivo, such biphasic dynamics could become cyclic, thus sustaining consecutive stages of metastatic spread. Although so relevant in cancer, the therapeutic targeting of cell migration has proven to be challenging and has given limited clinical success on cancer metastasis (Steeg, 2016). Our results suggest that a novel anti-metastatic strategy could be founded on harnessing the initial phases of migration by modulating the interaction between hERG1 and $\beta$1 integrin with the specific diabody scDb–hERG1–$\beta$1. This would not affect hERG1 function in other tissues, thus offering potential anti-metastatic effect without the cardiotoxic side effects that hERG1 blockers can exert (Duranti et al, 2021b; Santini et al, 2023).

# Materials and Methods

## Chemicals and antibodies

Unless otherwise, indicated chemicals were purchased from Sigma-Aldrich. Protein A/G Plus-Agarose for immunoprecipitation was from Santa Cruz Biotechnology (sc-2003). The following antibodies were used: mouse monoclonal antibody mAb hERG1 (MCK Therapeutics s.r.l.); the Alexa 488–conjugated mAB hERG1 was used at 1 $\mu$g/ml for FACS experiments; mouse monoclonal antibody (m-mAb) AKT1/2/3 (H-136) (cat. sc-8312; Santa Cruz Biotechnology) at a final dilution 1:500 for WB; m-mAb p-Akt1/2/3 (Thr 308) (cat. sc-271966; Santa Cruz Biotechnology) at a final dilution 1:500 for WB; r-pAb anti $\beta$1-integrin, RM-12 (Immunological Science) at final dilution 1:1,000 for WB; r-pAb anti-hERG1, C54 (MCK Therapeutics s.r.l.) at a final dilution 1:1,000 for WB; scFv–hERG1-Alexa-647 (MCK Therapeutics Srl) at 1:50 for IF; m-mAb $\beta$1 TS2/16 (Ultra-LEAF Purified anti-human CD29 Antibody, cat. 303035; Bio Legend) 5 $\mu$g antibody/mg protein for Co-IP, 1:500 for IF, 1:1,000 for cells pre-treatment; scFv–$\beta$1 integrin-Alexa-647 1:50 for IF m-mAb anti-$\alpha$-tubulin (cat. T9026; Sigma-Aldrich) at 1:500 dilution for WB; m-mAB anti KCNE1 at 1:500 for WB (Abcam), m-mAb anti-G$\alpha$i3 for WB (1:500) and IF (1:500) (Santa Cruz Biotechnology), m-mAb anti-girdin at 1:500 for WB and 1:200 for IF (Santa Cruz Biotechnology) and r-pAb anti-RAB 5-Alexa-488 at 1:500 for IF (Thermo Fisher Scientific). The scDb–hERG1/$\beta$1 (MCK Therapeutics s.r.l) was used either Alexa-647–conjugated on fixed cells for IF at the dilution 1:50 (addressed as "fluorescent scDb–hERG1–$\beta$1) or naked for treating live cells at 20 $\mu$g/ml (see below).

Secondary antibodies used for WBs were: anti-rabbit immunoglobulin G (IgG) peroxidase antibody (1:10,000; whole molecule, A0545), anti-mouse IgG peroxidase antibody (1:5,000; whole molecule, A4416), IRDYe 800 CW anti-mouse (1:20,000; LI-COR Biosciences), and IRDye 800CW anti-rabbit (1:20,000; LI-COR Biosciences) secondary antibodies. Alexa Fluor 488 goat anti-mouse antibody (Thermo Fisher Scientific) was used 1:500 and anti-6xHis antibody (Abcam) was used 1:250. Hoechst was used for staining nuclei in IF experiments (1:1,000 in PBS, 45 min; Merck Sigma).

Actinomycin D (Act), cytochalasin (Cyto-D), cycloheximide (Cyclo), proteinase K (PK), endoglycosidase H (Endo-H), and N-glycosidase F (N-Gly F) were purchased from Sigma-Aldrich and used at a concentration of 1 $\mu$g/ml, 1 $\mu$g/ml, 10 $\mu$g/ml, 100 $\mu$g/ml, 1 $\mu$M, and 1 $\mu$M, respectively.

Regarding pertussis toxin (PTX; Hello Bio Ltd), it was dissolved in ddH$_2$O at 100 $\mu$g/ml.

## Cells and culture

HEK293, PANC-1, and HCT116 cells were obtained from the American Type Culture Collection. Cells were routinely cultured at 37°C with 5% CO$_2$ in a humidified atmosphere, in RPMI (Euroclone) (HCT116) or in DMEM (Euroclone) (HEK293, PANC1), supplemented with 2% L-glut and 10% FBS (FBS EU Approved; Euroclone). We certify that all the cell lines used in the present study were routinely screened for mycoplasma contamination, and only mycoplasma-negative cells were used. HEK293 cells expressing the hERG1 constructs (HEK–hERG1, HEK–R531C, HEK–K525C) were prepared as previously described (Becchetti et al, 2017) and maintained in complete culture medium supplemented with 0.8 mg/ml of geneticin (G418; Thermo Fisher Scientific). Time zero is defined as the timepoint corresponding to the cells seeding.

## Preparation of cells for experiments

For all the experiments, all the cell types were starved overnight (O/N) by culturing them in serum-free BSA medium, that is, DMEM or RPMI 1640 for HCT116 cells, containing 250 $\mu$g/ml of heat-inactivated (HI) BSA (Fraction V; Euroclone). For PTX experiments, the toxin was added during the O/N incubation at the final concentration of 100 ng/ml. The day after, cells were harvested by detaching them with 5 mM EDTA in PBS and resuspended in serum-free medium. Next, cells were seeded on FN-coated dishes/slides. To this purpose, culture dishes/glass slides were first coated with fibronectin (human plasma; Sigma-Aldrich) diluted in sterile PBS (Euroclone) at 5 $\mu$g/cm$^2$ concentration. The culture surface was coated with a minimal volume. The dishes were left air-drying for 1 h at room temperature before introducing cells and medium. Hence, cells detached as above and resuspended in serum-free BSA medium, were added at different concentrations depending on the type of experiment to be performed (details are below). The following treatments were performed at this point: (1) treatment with TS2/16. The TS2/16 antibody was added at 1 $\mu$g/ml final concentration in the serum-free BSA medium and cells were then kept in suspension flasks for different time points before collecting them for time course experiments; (2) treatment with PTX. No further additions of the toxin were performed, after the O/N pre-incubation; (3) treatment with E4031. E4031 was added in the in the serum-free BSA medium at the final concentration of 40 $\mu$M; (4) treatment with scDb–hERG1–$\beta$1. scDb–hERG1–$\beta$1 was added in the in the serum-free BSA medium at the final concentration of 20 $\mu$g/ml.

## Cell adhesion test

Cells starved O/N and collected as above were seeded on FN in six-well plates (500,000 cells per well). Each well was divided into four quadrants by drawing a cross on the bottom. At different time

points, the medium was discarded and PBS 1X was added to the wells. A brightfield image was then taken using the inverted optical microscope EuroClone EVOS xl (AMG) at a 10X magnification in correspondence of the cross drawn at the bottom of the well. Wells were then washed once with PBS to eliminate non-adhered cells. A second image of the wells in correspondence of the cross drawn on the bottom was then taken post-wash. For each condition, cells were counted both at pre- and post-wash stage, and the percentage of remaining adherent cells after PBS wash was calculated.

### Patch clamp recording

Cells starved O/N and collected as above were seeded on FN-coated or BSA-coated 35 mm Petri dishes at $0.3/0.4 × 10^6$/ml in 1 ml of serum-free BSA medium/dish. For experiments with HEK–hERG1–Tr, HEK–KCNE1–hERG1–Tr only GFP-positive cells were considered for recordings. Electrophysiological recordings were performed at room temperature (~25°C) in the whole-cell configuration of the patch clamp technique, at different time points from cell seeding (T0, T5, T20, T45, T60, T90, T120, T180, T240, and T300) during which cells were maintained in an incubator at 37°C, 5% $CO_2$. The patch pipettes were pulled from borosilicate glass capillary tubes, their resistance was 4–5 MΩ and their capacitances were manually compensated after reaching a stable gigaseal. The cell capacitance and series resistance were compensated (up to 75–85%) before running voltage-clamp protocols. scDb–hERG1–$\beta$1 was diluted in serum-free BSA medium and incubated 90 min at 37°C, 5% $CO_2$ before recordings. Experimental protocols and data acquisition were performed with the Multiclamp 700A or Multiclamp 1D amplifiers, and pCLAMP 9.2 software (Molecular Devices) has been used for data analysis. The hERG1 inward tail currents were measured with a 25 KHz sampling rate and a 2 KHz low-pass filter. Cells' identification and patch was performed at 40x magnification with a Nikon Eclipse TE300 microscope (Nikon Instruments Inc.), equipped with a Photometrics CoolSNAP CF camera (Teledyne Photometrics). Cells membrane potentials were held at −80 mV and hERG1 inward tail currents were elicited using a preconditioning holding potential ranging from 0 mV to −100 mV (10 mV step increment) followed by 1 s hyperpolarizing step (−120 mV) with an intersweep interval of 15 s. The internal pipette solution for hERG1 conductance measurement contained (in mM): 130 $K^+$ aspartate, 10 NaCl, 4 $CaCl_2$, 2 $MgCl_2$, 10 HEPES–NaOH, 10 EGTA, pH 7.3. The external solution, instead, contained (in mM): 130 NaCl, 5 KCl, 2 $CaCl_2$, 2 $MgCl_2$, 10 HEPES, 5 glucose ($E_K$ = −80 mV), pH 7.4. Resting membrane potential ($V_{rest}$) values for all the cell lines were measured in I = 0 mode. $I_{hERG1}$ amplitude, normalized to the maximum current amplitude, was used to construct the activation and inactivation curves. The half-maximum activation and inactivation voltages ($V_{1/2}$) were calculated by fitting I/V curves with Boltzmann functions. T was calculated for each recording by fitting the traces evoking the maximal tail current with a double exponential function.

### Real-time PCR

Total RNA was extracted following the TRIzol reagent (Thermo Fisher Scientific) protocol. *hERG1* mRNA was quantified by quantitative real-time polymerase chain reaction (qRT-PCR), using the PRISM 7700 sequence detection system (Applied Biosystems) and

the SYBR Green PCR Master Mix Kit (Applied Biosystems) as in the study of Iorio et al (2022). The relative expression of hERG1 was calculated by 2^(−delta δ CT) method (Livak & Schmittgen, 2001). GAPDH housekeeping gene was used as standard reference.

Primers used were the following: GAPDH-F: 5′-AGACAGCCG CATCTTCTTGT-3′; GAPDH-R: 5′-CTTGCCGTGGGTAGAGTCAT-3′; hERG1-F 5′-ACGTCTCTCCCAACACCAAC-3′; hERG1-R 5′-GAGTACAGCCGCTGG ATGAT-3′

### Cell transfection

HEK293-WT and HEK293 cells stably expressing KCNE1 (HEK293–KCNE1) (obtained as the study of Becchetti et al [2017]) were routinely cultured in DMEM (Euroclone) supplemented with 10% FBS, 10% L-glutamine, and 0.8 mg/ml G418 at 37°C in 5% $CO_2$. HEK–hERG1–Tr (these are HEK 293 WT transiently transfected with hERG1), HEK–KCNE1–hERG1–Tr (these are cells stably transfected with KCNE1, which have been transiently transfected with hERG1). Twenty-four hours after plating, cells were transfected with pcDNA3.1/hERG1 and pcDNA3.1/GFP plasmids in a 3:1 ratio (Cherubini et al, 2005) using Lipofectamine 2000 reagent (Invitrogen, Thermo Fisher Scientific) in OptiMEM medium (Gibco, Thermo Fisher Scientific), according to manufacturer's instructions. Following 5 h from transfection, the medium was replaced. Cells were then used for experiments within 48 h.

### RNA silencing

The silencing of HEK293, PANC, and HCT cells was carried out using Lipofectamine 3000 transfection reagent (Thermo Fisher Scientific), following manufacturer's protocol, using a validated siRNA against girdin (Catalog # 4392420), along with a negative control siRNA (Catalog #4390843) (both from Invitrogen).

### Protein extraction, co-immunoprecipitation, and Western blotting

Cells starved O/N and collected as above were seeded on FN-coated or BSA-coated 100 mm Petri dishes at $1.5/1.75 × 10^6$ cells/ml in 5 ml of serum-free BSA medium. Immunoblotting and co-immunoprecipitation were performed as previously described in the study of Becchetti et al (2017). Adherent cells were first washed with ice-cold PBS and then collected by scraping. Pellets were obtained by centrifugation at 400*g*, washed twice in PBS and then immediately incubated for 20 min in 1% NP-40 lysis buffer (1% NP-40, 150 mM NaCl, 50 mM Tris–HCl, pH 8, 5 mM EDTA, 10 mM $Na_4P_2O_7$) supplemented with a tablet of a complete mix of protease inhibitors (Roche Complete Mini; Roche Diagnostics). All the procedures were performed maintaining samples on ice. Lysates were centrifuged at 13,000*g* for 10 min at 4°C. Supernatants were then collected and assayed for protein concentration using Bradford protein assay (Bio-Rad), following manufacturer's instructions. For co-immunoprecipitation, samples (1.5 mg of protein) were subjected to a pre-clearing step, consisting of 2 h incubation at 4°C under rotation with Protein A/G Plus-Agarose (Sigma-Aldrich) beads, following manufacturer's instructions. Thereafter, cell lysates were immunoprecipitated with TS2/16 antibody at the

concentration indicated in "Chemicals and antibodies" by overnight incubation at 4°C under gentle rotation. Beads were washed three times with PBS, and the bound protein component was finally eluted by boiling the samples in Laemmli buffer for 5 min at 95°C. The obtained samples were run on a 7.5% polyacrylamide gel for 1 h at 100 V for 1 h in Tris–glycine-SDS running buffer (Bio-Rad). Gels semi-dry blotting on PVDF membranes was performed with TurboBlot (Bio-Rad) using the "HIGH MW" program (1.3 A, 25 V for 10 min). Membranes were incubated 2 h at room temperature with 0.1% Tween 20 in PBS (T-phosphate–buffered saline) containing 5% BSA (T-phosphate–BSA). Blots were then incubated overnight at 4°C with polyclonal antibodies against hERG1 (C54) or $\beta$1 integrin (RM12) at the concentrations indicated in "Chemicals and antibodies."

Membranes were then washed three times with T-phosphate–buffered saline and incubated with horseradish-peroxidase anti-rabbit secondary antibodies for 45 min at room temperature at the concentration indicated in "Chemicals and antibodies." After three washes with T-phosphate–buffered saline, the immunoreactivity was determined by enhanced chemiluminescent reaction using ECL$^{TM}$ peroxidase substrate (GE Healthcare) and the ImageQuant$^{TM}$ LAS 4000 image capture system (GE Healthcare). For membranes stripping the ReBlot WB recycling kit (Merck Millipore) was routinely used according to manufacturer's instructions. For cell signaling experiments and co-immunoprecipitation inputs signal detection, proteins were extracted and quantified as previously described. Fifty $\mu$g of protein were assayed for pAkt/Akt, hERG1, $\beta$1, G$\alpha$i3, and KCNE1. Samples were denatured in 4X Laemmli buffer at 95°C for 5 min and then run by sodium dodecyl sulfate-poly-acrylamide gel electrophoresis (SDS–PAGE) under the previously described settings. 7.5% polyacrylamide gels were used for hERG1, $\beta$1-integrin, hERG1, G$\alpha$i3, and KCNE1, whereas 10% gels were used for pAkt/Akt analysis. Blotting was performed using the Turbo-Blot (Bio-Rad) "MIXED MW" program (1.3 A, 25 V for 7 min) for pAkt/Akt, whereas the "HIGH MW" program was selected for hERG1 and $\beta$1 integrin.

WB was performed using primary and secondary antibodies diluted in T-phosphate–BSA at the concentrations indicated in "Chemicals and antibodies." Membranes blocking and washing was performed as described for co-immunoprecipitation experiments. Immunoreactivity for hERG1 and $\beta$1 integrin was detected by chemiluminescence as previously described, whereas pAkt/Akt, G$\alpha$i3, and KCNE1 were revealed by using IRDYe 800 CW anti-mouse and anti-rabbit secondary antibody (concentrations reported in "Chemicals and antibodies") and the LI-COR Odyssey Scanner apparatus (LI-COR Biosciences).

### Densitometric analysis

Densitometric analysis was performed using ImageJ software (ImageJ v.1.38; U.S. National Institutes of Health) on three different scans after background subtraction. Results were obtained from at least three different independent experiments. For hERG1/$\beta$1 integrin complex quantification (Becchetti et al, 2017), the signal for the co-immunoprecipitated protein (hERG1) was divided by the signal of the protein used for immunoprecipitation ($\beta$1 integrin) and then normalized to the signal of the corresponding protein in the total lysate ($\beta$1 integrin input). The resulting value is indicated as "hERG1/$\beta$1 integrin complex" throughout the article and in the figures.

### IF

Cells starved O/N and collected as above were seeded on FN-coated or BSA-coated round glass coverslips inserted into a well of a 24-well clusters. The final concentration of the cells was $5 \times 10^4$ cells/well in 0.5 ml of serum-free BSA medium. IF on cells was performed by following the protocol previously described in the study of Duranti et al (2018, 2021a, 2021b). After 2 h of blocking in PBS with 10% BSA, sections were incubated for further 2 h with scDb–hERG1–$\beta$1 (20 $\mu$g/ml final concentrations) followed by 1 h with anti-6xHis (Abcam) and then 1 h with anti-mouse Alexa Fluor 488 (Thermo Fisher Scientific). Incubation with scFv–hERG1-Alexa-647, scFv–$\beta$1 integrin-Alexa-647, RAB5, G$\alpha$i3, and girdin antibody was performed O/N (see the "Chemicals and antibodies" section), secondary antibody incubation for 60 min in the dark was performed. Nuclei were stained with Hoechst and slides were mounted using ProLong Diamond Antifade Mountant (Invitrogen). To investigate cytoskeletal actin architecture using confocal microscopy, cells were fixed using 4% PFA, followed by permeabilization (0.1% Triton-X; Sigma-Aldrich), blocking with 1% BSA (Sigma-Aldrich), and staining with rhodamine-conjugated phalloidin, following manufacturer's instructions (Invitrogen) and Hoechst (see the "Chemicals and antibodies" section). All images were captured using a confocal microscope, Nikon Eclipse TE2000-U (Nikon).

### Quantification of total/plasma membrane/cytoplasmic IF signals

In all the IF experiments, the fluorescence relative to 20 cells (in 10 different fields and 3 different experiments) per each condition was determined using ImageJ software. For total IF signal, we considered the mean fluorescence intensity for each cell. When needed, the fluorescent intensity at the plasma membrane level was quantified considering exclusively the signals arising at the periphery of the cells. To this purpose, we draw a mask (highlighted by white circular lines in the figures) that was performed determined using the "freeform line profile" function drawn around the cell surface that enucleates only the fluorescent signal related to the membrane which was subsequently quantified using ImageJ software (Brackenbury et al, 2007). To quantify RAB5 fluorescence in the cytoplasm, we considered the signals arising exclusively in the cytoplasmic area below the plasma membrane, highlighted by the white mask as above. For all quantifications, data are reported as fold increase in a. u. (arbitrary unit) between either 0 and 2.5 or 0 and 3.0, as specified in the figure legends. Cortical actin density was quantified using the dedicated PlasMACC Fiji Plugin (Kurps et al, 2014). Co-localization was quantified using Manders' Overlap coefficient, which was measured using the Co-loc plugin (Fiji software) for each cell analyzed.

### Flow cytometry

Flow cytometry was performed to assess hERG1 and $\beta$1 expression pretreating cells 20' with FN or TS2/16 to stimulate hERG1/$\beta$1 complex formation. Cells were then revealed using TS2/16 and mAb hERG1-Alexa 488 using BD FACSCanto Flow Cytometer as in the study of Pillozzi et al (2011). PTX treatment was performed the day before the experiments. Acquisition and analysis were performed

using FACSDiva software (BD Biosciences). Values are expressed as mean fluorescence intensity of the area under the curve, indicated as mean florescence intensity.

### Lateral motility and haptotaxis

Lateral motility was determined using 35 mm dishes and drawing 15 horizontal lines and 3 perpendicular lines on the dish bottom, as in the study of Iorio et al (2020). Plates were coated with FN (human plasma; Sigma-Aldrich), and $5 \times 10^5$ cells were seeded and allowed to attach 5–10 min. Then, a manual scratch was carried out and the width of the wound was determined (W0). At this time, the different treatments were added. Then, dishes were incubated for further 90 min. At the end of incubation, the width of the wounds (Wt) was determined on unfixed cells. We took care to do these measurements within not more than 30 min overall. Motility index (MI) was assessed using the following formula: MI = 1 − Wt/W0, where Wt is the width of the wounds.

Haptotaxis experiments were performed according to the protocol described in the study of Leyme et al (2015). The bottom side of membrane filters (8 $\mu$m pores, 24-well format TC inserts; Sarstedt) was coated with FN (human plasma; Sigma-Aldrich). Cells were treated with scDb–hERG1–$\beta$1, PTX and silenced for girdin expression. Cells were allowed to migrate for 120 min at 37°C, before fixing and staining with crystal violet. Migrating cells at the bottom side of the filters were counted as follows: images were acquired with an inverted microscope (EVOS X; Advanced Microscopy Group) at a 40X magnification. Contrast phase pictures for 5 different optical fields were taken and cells were counted. Haptotaxis results are reported as the mean number of migrating cells in five optical fields acquired in triplicate experiments.

### Mathematical model

Some of the model parameters are fixed either through an order-of-magnitude estimate based on direct observational knowledge of the system (e.g., order of magnitude of *hERG1* mRNA increase) or on experimental data (e.g., approximate quantity of integrins in the system), whereas the other must be determined by adjusting the theoretical predictions on the experimental data (see supplementary information). An initial educated guess is obtained by *manually* exploring the parameter space. Then, a fine fitting of the parameters is carried out using a stochastic Monte-Carlo method. At each iteration, an array of test parameters is generated by slightly varying a random pair of parameters. For each new test vector of parameters, a score (cost function) is computed as a weighted sum of the squared deviations between the experimental data and the prediction of the model, obtained by numerically solving the set of rate equations. The algorithm is iterated by accepting new guesses of the floating parameter vector until the relative decrease of the score falls below a prescribed tolerance. A detailed description is reported in the Supplemental Data 1, file entitled "*Whole-cell network model of hERG1 synthesis, cytoplasmic trafficking and complex formation with integrins on the plasma membrane.*"

### Statistical analysis

Unless indicated otherwise, data are given as mean values ± SEM, with "n" indicating the number of independent experiments. At least three independent experiments were performed. Statistical comparisons were performed with OriginPro 2015 and SAS 9.2 (SAS Institute) software. The normality of data distribution was checked with K–S test. In the case of normal distributions, each data set was first checked for variance homogeneity, using the F test for equality of two variances and the Brown–Forsythe test for multiple comparisons. For data with unequal variances, the Welch correction was applied. For comparisons between two groups of data, we used the *t* test. A two-sample K–S test was performed to test whether two underlying probability distributions differed. For multiple comparisons, one-way ANOVA followed by Bonferroni's post hoc test was performed to derive *P*-values.

## Data Availability

Data are available upon request.

## Supplementary Information

## Acknowledgements

This research was funded by the University of Florence (ex 60%) to A Arcangeli. This work was supported by Associazione Italiana per la Ricerca sul Cancro (AIRC, grant nos. 1662, 15627, and 21510) to A Arcangeli, PRIN Italian Ministry of University and Research (MIUR) "Leveraging basic knowledge of ion channel network in cancer for innovative therapeutic strategies (LI-ONESS)" 20174TB8KW to A Arcangeli, pHioniC: European Union's Horizon 2020 grant No 813834 to A Arcangeli. J Iorio was supported by Regione Tocana fellowship within the project "Progetti di alta formazione attraverso l'attivazione di Assegni di Ricerca" (MutCoP project) co-funded by Fondazione Cassa di Risparmio di Pistoia e Pescia and was formely funded by a fellowship of Fondazione Cassa di Risparmio di Pistoia e Pescia within Giovani@Ricerca Scientifica program. This work was also supported by the University of Milano-Bicocca to A Becchetti (grant 2021-ATE-0042). C Duranti was supported by an AIRC fellowship for Italy "Francesco Tonni" ID 24020. The data presented in the current study were in part generated using grants by the European Union - NextGenerationEU - National Recovery and Resilience Plan, Mission 4 Component 2 - Investment 1.5 - THE - Tuscany Health Ecosystem - ECS00000017 - CUP B83C22003920001 to C Duranti and A Arcangeli. The data presented in the current study were in part generated using grants by European Union, National Recovery and Resilience Plan, Mission 4 Component 2 – Investment 1.4 - National Center for Gene Therapy and Drugs based on RNA Technology - NextGenerationEU – Project Code CN00000041-CUP B13C22001010001 to A Arcangeli. We thank Massimo D'Amico, Luca Gasparoli, Sagar Manoli, Silvia Crescioli, Serena Pillozzi, and Olivia Crociani for producing some preliminary data. We thank Rossella Colasurdo for the help in maintaining PANC-1 cells during revision experiments.

## Author Contributions

C Duranti and J Iorio: data curation, formal analysis, validation, investigation, visualization, methodology, and writing—original draft, review, and editing.

G Bagni and G Chioccioli Altadonna: data curation, formal analysis, investigation, and methodology.

T Fillion: formal analysis and methodology.

M Lulli, FN D'Alessandro, and A Montalbano: investigation and methodology.

E Lastraioli: methodology and writing—review and editing.

D Fanelli and T Schmidt: supervision and methodology.

S Coppola: methodology.

F Piazza: conceptualization, supervision, methodology, and writing—original draft, review, and editing.

A Becchetti: conceptualization, supervision, project administration, and writing—original draft, review, and editing.

A Arcangeli: conceptualization, supervision, funding acquisition, project administration, and writing—original draft, review, and editing.

## Conflict of Interest Statement

Annarosa Arcangeli is founder and partner of the University of Florence spin-off MCK Therapeutics s.r.l. All the other authors declare no conflict of interest.

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
