## [Reviewer comments · Life Science Alliance]

Life Science Alliance

Integrins regulate hERG1 dynamics by girdin-dependent G α i3: signaling and modeling in cancer cells

Claudia Duranti, Jessica Iorio, Giacomo Bagni, Ginevra Chioccioli Altadonna, Thibault Fillion, Matteo Lulli, Franco D'Alessandro, Alberto Montalbano, Elena Lastraioli, Duccio Fanelli, Stefano Coppola, Thomas Schmidt, Francesco Piazza, Andrea Becchetti, and Annarosa Arcangeli

DOI: <https://doi.org/10.26508/lsa.202302135>

Corresponding author(s): Annarosa Arcangeli, University of Florence

Review Timeline:

Submission Date:	2023-05-06
Editorial Decision:	2023-06-16
Revision Received:	2023-09-08
Editorial Decision:	2023-10-04
Revision Received:	2023-10-13
Editorial Decision:	2023-10-20
Revision Received:	2023-10-22
Accepted:	2023-10-23

Transaction Report:

June 16, 2023

Re: Life Science Alliance manuscript #LSA-2023-02135-T

Prof. Annarosa Arcangeli
University of Florence
Experimental and Clinical Medicine and Istituto Toscano Tumori
Viale Morgagni 50
Firenze 50134
Italy

Dear Dr. Arcangeli,

Thank you for submitting your manuscript entitled "Integrins regulate hERG1 dynamics by girdin-dependent Gαi3: signaling and modeling in cancer cells" to Life Science Alliance. The manuscript was assessed by expert reviewers, whose comments are appended to this letter. We invite you to submit a revised manuscript addressing the Reviewer comments.

Thank you for this interesting contribution to Life Science Alliance. We are looking forward to receiving your revised manuscript.

Sincerely,

B. MANUSCRIPT ORGANIZATION AND FORMATTING:

Reviewer #1 (Comments to the Authors (Required)):

The work presented by Claudia Duranti et al. compiles a fascinating manuscript that sheds light on the complex signaling of cancerous cells. The authors present a series of observations that aim to elucidate the signaling pathway that involve the b1 integrin-hERG complex that only happens in the context of cancer. hERG potassium channel is one of the most exciting potassium channels, it plays critical roles in cardiac physiology, and interestingly, it is dramatically overexpressed in tumors and during neoplastic progression. In cancer cells, hERG associates with integrins, forming a unique complex (b1-integrin/hERG). Integrins are membrane proteins that form macro complexes relevant to metastatic dissemination.

The authors present a well-written and extensive body of work that explores the pathway downstream of the b1-integrin/hERG complex to elucidate the reasons for aberrant hERG expression on cancer pathophysiology and cell motility dynamics. Their approach combines functional experiments and a mathematical model to help explain hERG overexpression and cell migration in cancer cells. The authors describe a pathway downstream b1 integrin/hERG complex that regulates f-actin proteins involved in cell motility and migration in two phases; a fast one that involves hERG expression and a slow one that seems to involve non-functional channels and endocytosis. They use functional studies such as Electrophysiology, WB, immunofluorescence, flow cytometry, specific blockers, and antibodies to test the implication of the b1-integrin/hERG complex, GEF-girdin, G protein and Akt, f-actin, etc. on the pathway. However, the current version lacks the direct or indirect nature of the molecular interactions between the b1-integrin/hERG complex and the downstream elements such as girdin or G proteins. However, I understand that this precise interaction might not be the main focus of this manuscript, and addressing it will require extra work; that said, the pathway unveiled in the present manuscript shows valuable and exciting evidence for cancer pathophysiology. My suggestions are to help expand some sections that will help clarify and reinforce the story.

MAJOR POINTS

1. The evidence suggests that forming the hERG-b1 complex in the membrane during cell adhesion triggers the pathway. However, it's not clear which is the first activator of the integrin or the trigger of the b1-integrin with the hERG channel in the membrane; in lines 183 and 193, the authors mention fibronectin (FN) as a stimulator of hERG expression and translocation, is that the author's interpretation? Please clarify.
2. One of the fascinating features of the b1-integrin/hERG complex is the state-dependent of its formation. Previous work by the authors shows how the association with integrins doesn't depend on the ability of the channel to conduct ions, but b1-integrin prefers the closed state of the channel to form the complex. However, It needs to be clarified what's exactly the role of the hERG channel in the signaling pathway. In the present study, there's an opportunity to address the role of hERG in triggering and regulating the biphasic effect and the downstream signaling path. There are some approximations to address hERG's role by testing KCNE1 to inhibit the b1-integrin complex and two mutants that stabilize the channel in different states. However, it's not discussed whether hERG needs to function as a channel or its role in this scenario is as a metabotropic receptor triggering by itself the GEF girdin and Gai downstream. The increase in current density (t90) and the changes in membrane potential suggest that this is the case, but the currents are not normalized (see minor point 1), so it's hard to compare current densities in different conditions. Please clarify this point. To address this specific point, I suggest using a channel blocker such as E403 or BeKm-1 toxin and checking the elements downstream, such as girdin, G protein levels, and cell motility.
3. The results presented in Fig 2A, showing increasing levels of hERG mRNA from t0 to t300, are not clear to me. If I interpret the data correctly, it is clear that after t90, the protein level decreases, and the effect of hERG on the membrane potential is also visible (Fig1). However, those mRNA levels do not translate into protein. Do the authors have an explanation of what happened? In the mathematical model, the authors assume degradation after synthesis, but the experimental evidence presented in Fig2A doesn't agree. Please clarify.
4. The final and most exciting part of the present manuscript is the f-actin re-organization which is the final goal of the study. It seems sensible to the blockers of elements in the signaling pathway; however, does blocking the hERG potassium channel affect the f-actin like the other blockers do? (see minor point 5)

MINOR POINTS

1. In my opinion, presenting the currents densities (like in Fig 1) on absolute peak current could be misleading since the absolute values will depend on many factors, such as cell size, therefore, to correct for the size of the cell, I will suggest normalizing the absolute current by the size of the cell recorded (pA/pF).
2. Fig 5 A, the experiments shown were measured at t45. Is there a reason for that? The optimal time for the complex is t90; please state how do the data look at that time point?
3. In Line 270, the authors state: "We conclude that assembly of the hERG1/b1 integrin complex drives the normal hERG1 expression/translocation process" What do the authors mean by normal? If I understood correctly, this complex was just present in cancer cells.
4. If I understood correctly, the scDb-hERG1-b1 could be used to image the hERG-b1 integrin (ine284, 297) but also to disrupt the complex formation between herg1 and b1-integrin (line267 and Fig 5F) at t90. It gets confusing since it's used both ways along the figures, please clarify in the main text.
5. Fig 10 G, the experiments shown were measured at t120. Is there a reason for that? Looking at the other panels within the figure one will expect to see the biggest effect at t90. How do the data look like at t90 for these experiments? Please state.
6. In Line 466, the authors state that: "...the hERG localization peak on the plasma membrane to relax more quickly in cancer cells." maybe I missed it, but I didn't find the experimental data to support this. I will suggest fitting the data with an exponential equation (single or double) to assess the time constant of the hERG localization on the plasma membrane.

TYPOS

line 186 nonreceptor, no dash and line 196 non-receptor with dash. Please unify
Fig 5 panel B no label

Reviewer #2 (Comments to the Authors (Required)):

Duranti, Iorio and their colleagues study the dynamics of hERG1 and its interaction with integrin in the context of cancer cell migration. Authors first show how hERG1 expression and its plasma membrane localization depend on integrin activation. They further investigate the role of Galphai3 interaction with integrin. The hERG1/integrin complex formation and how it regulates the translocation of hERG1 to the plasma membrane are also studied in detail. In addition, authors use two different cell lines to test whether the mechanisms they observed in their previous experiments valid in the context of pancreatic and colorectal cancers. Authors then develop a mathematical model for the integrin dependent hERG1 dynamics, which seem to recapitulate the experiments reasonably well. Finally, they investigate the effect of hERG1/integrin complex on the actin network organization.

The manuscript contains significant amount of experimental data supporting the mechanism suggested in the manuscript. The main text is well-written and the data in Figures 1-5 is explained in detail including interpretation. For the rest of the figures, authors sometimes describe the data only (i.e. their observations), rather than including detailed insights on the mechanism as they did in the previous figures. That being said, authors include a lengthy discussion section, which summarizes their findings well. The discussion mostly focuses on the interpretation of the mathematical model, which does not have detailed explanation in the manuscript (see first comment below). Overall, I think this work provides detailed investigation on the underlying mechanisms of hERG1 dynamics. My comments below might be useful to improve the manuscript.

Comments:

- My biggest concern is about the description of the mathematical model. Authors refer to the "appendix" for the model details, however, there is no appendix provided. If they refer to the Materials and Methods section as an appendix, the "Mathematical model" section does not provide enough details. Authors mention that they varied the parameters, but how time course is modeled is not explained. Fig8C shows the best values for the parameters used, but how each reaction takes place is not explained. Furthermore, it will be better to add the short descriptions of these parameters in the table, along within the detailed description of the model. Nevertheless, based on Fig8A, the model seems reasonable. A detailed description of the model is needed in the materials and methods.
- In Fig1E, errors for the I and V (independent of Fig1F) and the fitting confidence intervals would be useful.
- In Fig2C (and the following figures), it is not clear how white masks are created. Please explain in more detail.
- Regarding Fig2D-G: Instead of using mean fluorescent intensity, why don't the authors use the total intensity (area under the curve)?
- Fig3A right panel should have the statistical test between T0 and PTX as for IhERG1 and Vrest, rather than CTR and PTX.

Looking at the errors on the bars, it is likely there will be a significant increase in the mean intensity. What would that mean?

- Fig3C colocalization images are not clear. Please write in caption which color represents what. Although there is a quantification of IF intensity below, is there an additional quantification specifically for colocalization, such as done for Fig4D-E-F? The caption is not very clear on how the intensity quantification is done. The same questions apply for Fig3 E-F-G.
- Fig5B label is missing, and the organization is not matching the caption. Overall, for each figure, it would be better in my opinion to give every piece of graph/image its own letter, instead of writing, for example: Fig5B left, Fig5B middle, Fig5B right, Fig5B bottom etc. Or other subindices such as 5Bi, 5Bii, 5Biii, 5Biv etc could be used if authors want to keep the same grouping.
- Fig5C left panel should also have statistical tests for each cell line between T0 and T300 in order to support the authors' claim that hERG1 mRNA expression increases for HEK-hERG1 cells but does not increase for HEK-hERG1-K525C cells.
- Fig5F bar graph should have a statistical test between T0 and scDb-hERG1-beta1, even though it seems like it will be non-significant (based on the error on the bars). More importantly, why there is no separate T0 data for CTR and scDb-hERG1-beta1, so that the authors do direct statistical tests between T0 and T90 for each condition? Same comments/concerns apply for Fig5G bar chart. I also wonder for 5F, how authors would comment on the shift in intensity with scDb-hERG1 compared to T0, if it's not translocating to plasma membrane as in the control.
- For Fig6, authors mention slow decay after T90. It is not clear what the "slow" is compared to? Is it compared to the rate of increase between T0 and T90? Is there any quantification on the decay rates? For example, in 6E the decay rate between T90 and T120 is much faster even when compared to the increase between T0 and T90 for both cell lines.
- Is this supposed to be Fig 7F right panels for the following statement: "hERG1/ β 1 integrin complex assembly evidenced by IF staining with scDb-hERG1- β 1 (Fig. 7G, right panels)."
- In Fig10B, please show n.s. for pERK to support the text. Another suggestion is to fit a line for pERK data in 10A and check whether the slope is different than zero or not (even though it's clear that the values stay relatively unchanged for pERK compared to pAKT).
- In Fig10C, T15 vs T90 is shown, but the main text mentions T0. Is there a typo in any of them (probably in the main text)?
- Please indicate which data is shown in Figs10E&F (i.e. the label for the vertical axes). Is the red data for actin length and is the green data for actin density? If so, please annotate the axis properly: For example, double vertical axes can be used (i.e. left is actin length and right is actin density, labeled with appropriate colors), or, as in the previous one, two graphs could be used.

Dear Editor,

Attached please find the revised version of the manuscript entitled "**Integrins regulate hERG1 dynamics by girdin-dependent $G\alpha i3$: signaling and modeling in cancer cells**".

We have revised the manuscript according to Reviewers' comments. In particular:

-we have added a new Table 1, where data present on panel E of Figure 1 of the previous version are shown, in addition with current density (I_{dens}) data.

-the details of the mathematical model are now provided as a Supplemental Material pdf file entitled "*Whole-cell network model of hERG1 synthesis, cytoplasmic trafficking and complex formation with integrins on the plasma membrane*"

-the best-fit values of the floating parameters of the model have been moved from panel C of Figure 8 (which is now the new Figure 9) to Table 2, for clarity of the reader.

-the main experimental addition we have performed is represented by new experiments to address Reviewer's #1 comments 2 and 4 (see below, in the point by point answers to Reviewers' comments). In particular we have treated cancer cells with the hERG1 open channel blocker E4031 and with the diabody specific for the hERG1/ $\beta 1$ integrin complex, scDb-hERG1- $\beta 1$. The results of these experiments are reported in the new Figures 8 and 11.

All the changes made to the text are **highlighted in red and bold**.

Following is the point by point answers to the Reviewers' comments:

Reviewer #1

My suggestions are to help expand some sections that will help clarify and reinforce the story.

1. The evidence suggests that forming the hERG- $\beta 1$ complex in the membrane during cell adhesion triggers the pathway. However, it's not clear which is the first activator of the integrin or the trigger of the $\beta 1$ -integrin with the hERG channel in the membrane; in lines 183 and 193, the authors mention fibronectin (FN) as a stimulator of hERG expression and translocation, is that the author's interpretation? Please clarify.

As the Reviewer suggests, fibronectin (FN) is indeed the main activator of $\beta 1$ integrins in the cellular models studied in the present manuscript, as our group has previously demonstrated in the past (Arcangeli et al., 1993, Hofmann et al., 2001, Becchetti et al., 2017). We have clarified this concept in the Abstract, in the Introduction and in the Results sections. In other systems (i.e. human neuroblastoma cells) laminin also behaves as a $\beta 1$ integrin activator (Chioccioli Altadonna G et al., 2022). Finally, the TS2/16 monoclonal antibody is a specific activator of $\beta 1$ integrins, also in the absence of cell adhesion.

2. One of the fascinating features of the $\beta 1$ -integrin/hERG complex is the state-dependent of its formation. Previous work by the authors shows how the association with integrins doesn't depend on the ability of the channel to conduct ions, but $\beta 1$ -integrin prefers the closed state of the channel to form the complex. However, It needs to be clarified what's exactly the role of the hERG channel in the signaling pathway. To address this specific point, I suggest using a channel blocker

such as E403 or BeKm-1 toxin and checking the elements downstream, such as girdin, G protein levels, and cell motility.

We have performed the requested experiments with E4031, comparing its effects with those of scDb-hERG1- β 1, which has been proven in Duranti C. et al., 2021 to harness and then disrupt the hERG1/ β 1 integrin complex. Results are shown in the new figures 8 and 11 and described on pages 11, 13 and 14 of the Results section. Briefly, both molecules impaired the complex formation and the downstream signaling (G protein, girdin, f-actin organization, motility and haptotaxis) although scDb was even more effective than E4031, without blocking hERG1 currents (Figure 8A). These results suggest that the role of hERG1 in the signaling pathway downstream to integrin activation in cancer cells is mainly related to the promotion of hERG1/ β 1 integrin complex formation. Such interpretation is suggested by the fact that the complex formation is hampered by scDb-hERG1- β 1, which disrupts the macromolecular complex (Duranti et al., 2021) without blocking the current (Fig.8A), as well as by E4031, which impairs the complex formation by blocking the channel in the open state (Becchetti et al., 2017). This interpretation is consistent with the results obtained with the K525C hERG1 mutant in HEK transfected cells (Fig.5). Nonetheless, a rigorous definition of the role of hERG1 current in ECM activated signals will require further structural and functional studies in cancer cells. This reasoning has been reported in the Discussion, page 16.

3. The results presented in Fig 2A, showing increasing levels of hERG mRNA from t_0 to t_{300} , are not clear to me. If I interpret the data correctly, it is clear that after t_{90} , the protein level decreases, and the effect of hERG on the membrane potential is also visible (Fig1). However, those mRNA levels do not translate into protein. Do the authors have an explanation of what happened? In the mathematical model, the authors assume degradation after synthesis, but the experimental evidence presented in Fig2A doesn't agree. Please clarify.

mRNA levels increase with time as the signal triggered by active integrins (in all their forms, free and in complex with hERG1) kicks in, and reach a "plateau" level after T_{90} . This is not accompanied by an increase of the protein but, on the contrary, by a decrease, since the hERG1 protein is actively degraded through the Rab5 mediated pathway and other pathways (Fig.4D). More specifically, we find that, in order to fit the data, the signal triggered by activation of the integrins should also result in a slow-down of the degradation rate associated with the Rab5 mediated pathway (as demonstrated by the trend of the colocalization data with Rab5), while other degradation channels (represented collectively by the rate k_{dm} in the model) are assumed to be insensitive to the signal triggered by integrin activation. So, production of hERG1 mRNA and protein synthesis are boosted by the integrin activation signal (although the former is seen to proceed more slowly), alongside with trafficking and translocation. Degradation eventually causes the overall quantity of protein to decrease and reach a new stationary level that is characteristic of the fibronectin-activated dynamics. However, the difference between the dynamics of hERG1 and that of mRNA is also largely due to the non-linear dynamics of the activator-inhibitor trigger over time, in the presence of active integrin. In the short term, when the activator dominates, there is a boost in translation and trafficking, which disappears in the long term when the inhibitor dominates. In summary, in addition to active degradation, there is also non-linear regulation of synthesis and trafficking, which helps set up the hERG1 peak at T_{90} . A phrase in the Discussion (page 17) has been added to better clarify this point.

4. The final and most exciting part of the present manuscript is the f-actin re-organization which is the final goal of the study. It seems sensible to blockers of elements in the signaling pathway;

however, does blocking the hERG potassium channel affect the f-actin like the other blockers do? (see minor point 5)

We have performed new f-actin organization analysis with E4031. E4031 slightly increased stress fibers length, and decreased cortical f-actin density in PANC-1 and HCT 116 cells, at a lower extent compared to scDb and other blockers of the signaling pathway (new Fig. 11E and F). Consistently, E4031 treatment decreased cancer cell motility and haptotaxis, although to a lower extent compared to scDb and other blockers of the signaling pathway (new Fig. 11G and H). These results confirm our interpretation (see answer to question 2).

MINOR POINTS

1. In my opinion, presenting the currents densities (like in Fig 1) on absolute peak current could be misleading since the absolute values will depend on many factors, such as cell size, therefore, to correct for the size of the cell, I will suggest normalizing the absolute current by the size of the cell recorded (pA/pF).

According to Reviewer's suggestion, we now report the current density (pA/pF) values, in a new column of the new Table 1.

2. Fig 5 A, the experiments shown were measured at t45. Is there a reason for that? The optimal time for the complex is t90; please state how do the data look at that time point?

We apologize for the typo. The experiments were measured at T_{90} . We have amended the figure accordingly.

3. In Line 270, the authors state: "We conclude that assembly of the hERG1/b1 integrin complex drives the normal hERG1 expression/translocation process" What do the authors mean by normal? If I understood correctly, this complex was just present in cancer cells.

As the Reviewer correctly states, the complex is present only in cancer cells. We have erased the word "normal" from the text.

4. If I understood correctly, the scDb-hERG1-b1 could be used to image the hERG-b1 integrin (line 284, 297) but also to disrupt the complex formation between hERG1 and b1-integrin (line 267 and Fig 5F) at t90. It gets confusing since it's used both ways along the figures, please clarify in the main text.

We have clarified in the text Materials and Methods, Results and Figure legends) the difference, distinguishing between "*staining performed with fluorescent scDb-hERG1- β 1*" and "*treatment with scDb-hERG1- β 1*".

5. Fig 10 G, the experiments shown were measured at t120. Is there a reason for that? Looking at the other panels within the figure one will expect to see the biggest effect at t90. How do the data look like at t90 for these experiments? Please state.

We have better detailed the protocol used to determine the motility index in the Materials and Methods section (page 26). Indeed, the time scale for this type of experiment is slightly different

from that used in other experiments (i.e. we must consider the time of seeding before performing the scratch etc). Hence, we have erased the indication (T_{120}) from the previous Fig 10, in the new Figure 11.

6. In Line 466, the authors state that: "...the hERG localization peak on the plasma membrane to relax more quickly in cancer cells." maybe I missed it, but I didn't find the experimental data to support this. I will suggest fitting the data with an exponential equation (single or double) to assess the time constant of the hERG localization on the plasma membrane.

We have rephrased the sentence, which is now "As in HEK-hERG1 cells, PANC-1 and HCT116 cell adhesion on FN stimulated hERG1 localization in the plasma membrane, with a peak at T_{90} , which was followed by a comparatively slower decrease (Fig. 6C). A similar time course was observed in IF experiments with the anti-hERG1 scFv (Fig. 6D). The hERG1/ β 1 complex formation, determined by either co-IP experiments (Fig. 6E) or IF with fluorescent scDb-hERG1- β 1 (Fig. 6F), also showed a progressive increase in both cell types seeded on FN up to T_{90} , followed by a slower decay. The kinetics of these processes is quantified by the mathematical model described later."

The mathematical model used to fit the experimental data is described in the new Supplemental material, and the fittings are in the new Figure 9.

TYPOS

line 186 nonreceptor, no dash and line 196 non-receptor with dash. Please unify.

We have unified, as suggested.

Fig 5 panel B no label

Figure 5 and the labels of the different panels have been redrawn to accomplish Reviewer#2 requests.

Reviewer #2

- My biggest concern is about the description of the mathematical model. Authors refer to the "appendix" for the model details, however, there is no appendix provided..... A detailed description of the model is needed in the materials and methods.

We apologize for the mistake. The details of the mathematical model are now provided as a Supplemental Material pdf file entitled "*Whole-cell network model of hERG1 synthesis, cytoplasmic trafficking and complex formation with integrins on the plasma membrane*". For clarity to the reader, the file also contains panels relative to Fig.9 and Table 2.

- In Fig1E, errors for the I and V (independent of Fig1F) and the fitting confidence intervals would be useful.

Data in Figure 1E refer to single cells, hence we have not shown the errors. The fitting confidence intervals are now provided in the Figure legend.

- In Fig2C (and the following figures), it is not clear how white masks are created. Please explain in more detail.

We have added more details in the “Materials and methods” section. In particular, we have added a new paragraph: **Quantification of total/plasma membrane/cytoplasmic IF signals**. A new reference (Brackenbury W et al., 2007) has also been added.

- Regarding Fig2D-G: Instead of using mean fluorescent intensity, why don't the authors use the total intensity (area under the curve)?

We have indicated the mean fluorescent intensity meaning the “**mean fluorescent intensity of the area under the curve, MFI**”. In the revised version of the manuscript, we have specified better this point in the “Materials and methods” section as well as in the figure legends.

- Fig3A right panel should have the statistical test between T0 and PTX as for hHERG1 and Vrest, rather than CTR and PTX. Looking at the errors on the bars, it is likely there will be a significant increase in the mean intensity. What would that mean?

- Fig3C colocalization images are not clear. Please write in caption which color represents what. Although there is a quantification of IF intensity below, is there an additional quantification specifically for colocalization, such as done for Fig4D-E-F? The caption is not very clear on how the intensity quantification is done. The same questions apply for Fig3 E-F-G.

Figure 3 has been redrawn and its legend to accomplish Reviewer's requests. Each panel has now its own letter for better identification, with the only exceptions of panels E and G, where different images are included and indicated separately in the figure legend. Fig.3A contains the whole set of the statistical analysis. The latter has shown that the difference between T0 and PTX is not significant. We have reported the label NS on the graph. Color representation have been clarified, and a new paragraph in the “Materials and methods” section has been added (“**Quantification of total/plasma membrane/cytoplasmic IF signals.**”) to better explain how the IF signals and the co-localizations have been quantified.

- Fig5B label is missing, and the organization is not matching the caption. Overall, for each figure, it would be better in my opinion to give every piece of graph/image its own letter, instead of writing, for example: Fig5B left, Fig5B middle, Fig5B right, Fig5B bottom etc. Or other subindices such as 5Bi, 5Bii, 5Biii, 5Biv etc could be used if authors want to keep the same grouping.

- Fig5C left panel should also have statistical tests for each cell line between T0 and T300 in order to support the authors' claim that hHERG1 mRNA expression increases for HEK-hERG1 cells but does not increase for HEK-hERG1-K525C cells.

- Fig5F bar graph should have a statistical test between T0 and scDb-hERG1-beta1, even though it seems like it will be non-significant (based on the error on the bars). More importantly, why there is no separate T0 data for CTR and scDb-hERG1-beta1, so that the authors do direct statistical tests between T0 and T90 for each condition? Same comments/concerns apply for Fig5G bar chart. I also wonder for 5F, how authors would comment on the shift in intensity with scDb-hERG1 compared to T0, if it's not translocating to plasma membrane as in the control.

Figure 5 has been deeply redrawn to accomplish Reviewer's requests. Each panel has now its own letter for better identification. The statistical analysis has been modified according to Reviewer's suggestions. Moreover, we would like to clarify that T0 represents the cells before the seeding, as detailed in figure 1 legend.

- For Fig6, authors mention slow decay after T90. It is not clear what the "slow" is compared to? Is it compared to the rate of increase between T0 and T90? Is there any quantification on the decay rates? For example, in 6E the decay rate between T90 and T120 is much faster even when compared to the increase between T0 and T90 for both cell lines.

The text has been revised according to the Reviewer suggestion, making it clearer that the decay is slow as compared to the rising phase. We also explain that the decay rate is quantified by the mathematical model. We have hence rephrased the sentence at page 10, which is now "As in HEK-hERG1 cells, PANC-1 and HCT116 cell adhesion on FN stimulated hERG1 localization in the plasma membrane, with a peak at T₉₀, which was followed by a comparatively slower decrease (Fig. 6C). A similar time course was observed in IF experiments with the anti-hERG1 scFv (Fig. 6D). The hERG1/β1 complex formation, determined by either co-IP experiments (Fig. 6E) or IF with fluorescent scDb-hERG1-β1 (Fig. 6F), also showed a progressive increase in both cell types seeded on FN up to T₉₀, followed by a slower decay. The kinetics of these processes is quantified by the mathematical model described later."

- Is this supposed to be Fig 7F right panels for the following statement: "hERG1/β1 integrin complex assembly evidenced by IF staining with scDb-hERG1-β1 (Fig. 7G, right panels)."

Figure 7 has been modified, and some panels have been moved to the new Figure 8. Fig.7G now shows the data relative to IF using the fluorescent scDb-hERG1-β1. We have changed the text accordingly.

- In Fig10B, please show n.s. for pERK to support the text. Another suggestion is to fit a line for pERK data in 10A and check whether the slope is different than zero or not (even though it's clear that the values stay relatively unchanged for pERK compared to pAKT).

In the revised version of the manuscript, figure 10 has been changed into figure 11. As suggested, we have indicated "NS" label in panel B. The slopes of the lines fitting the experimental points requested by the Reviewer are in the figure legend.

- In Fig10C, T15 vs T90 is shown, but the main text mentions T0. Is there a typo in any of them (probably in the main text)?

We confirm that it was a typo and we have amended in text accordingly.

- Please indicate which data is shown in Figs10E&F (i.e. the label for the vertical axes). Is the red data for actin length and is the green data for actin density? If so, please annotate the axis properly: For example, double vertical axes can be used (i.e. left is actin length and right is actin density, labeled with appropriate colors), or, as in the previous one, two graphs could be used.

In the revised version of the manuscript, figure 10 has been changed into the new figure 11, which is deeply modified, including the re-labelling of vertical axes.

October 4, 2023

Re: Life Science Alliance manuscript #LSA-2023-02135-TR

Prof. Annarosa Arcangeli
University of Florence
Department of Experimental and Clinical Medicine, Section of Internal Medicine, Viale Morgagni 50
Firenze 50134
Italy

Dear Dr. Arcangeli,

Thank you for submitting your revised manuscript entitled "Integrins regulate hERG1 dynamics by girdin-dependent G α i3: signaling and modeling in cancer cells" to Life Science Alliance. The manuscript has been seen by the original reviewers whose comments are appended below. While the reviewers provide minor comments, which are below and should be addressed, we have found numerous issues with the provided figures during our data integrity check.

I will list the flagged concerns here. We will not publish this manuscript unless these are addressed. For the blots in question, providing additional replicates as Source Data will be helpful. In general, samples can not be properly compared if they are run on separate gels. Control blots can not be re-used between samples.

Figure Checks:

- Figure 4A, the T-15 ERG column does not look like the blot provided as Source Data
- Figure 4B, for columns T-0 and T-90, the boxes indicating the areas zoomed in could be better fit to match
- Figures S2A 3rd panel and S6D 1st panel are duplicates
- Duplicate blots in Fig S4A Tubulin row, T5/T15 and T45/T60. The source data blot labeling does not match the figure
- the loading controls for Fig S6C 1st panel is duplicated in Figure S11A Panc-1
- The last panel in Figure S7C is the same as the last panel in Figure S7D
- please add a scale bar to Figure S11C

To upload the revised version of your manuscript, please log in to your account: <https://lsa.msubmit.net/cgi-bin/main.plex>
You will be guided to complete the submission of your revised manuscript and to fill in all necessary information.

- A letter addressing the reviewers' comments point by point.
- An editable version of the final text (.DOC or .DOCX) is needed for copyediting (no PDFs).
- High-resolution figure, supplementary figure and video files uploaded as individual files: See our detailed guidelines for preparing your production-ready images, <https://www.life-science-alliance.org/authors>
- Summary blurb (enter in submission system): A short text summarizing in a single sentence the study (max. 200 characters including spaces). This text is used in conjunction with the titles of papers, hence should be informative and complementary to the title and running title. It should describe the context and significance of the findings for a general readership; it should be written in the present tense and refer to the work in the third person. Author names should not be mentioned.

B. MANUSCRIPT ORGANIZATION AND FORMATTING:

Sincerely,

Reviewer #1 (Comments to the Authors (Required)):

I would like to thank the authors for taking into consideration my comments and add new results to the study. I appreciate the effort on making additional experiments. The present version of the manuscript shows in my opinion a more complete story highlighting the pathway of activation of hERG/b1 integrins and hinting their role in cancer cell migration.

I just have a minor thing, Fig 8, panel A and B : the labels of the X axes of the bar graphs are missing. Does the color code shown in Panel B (black/green/red) apply to the whole figure? Otherwise please state.

Other than that, I have no further comments.

Reviewer #2 (Comments to the Authors (Required)):

Duranti, Iorio and their colleagues updated their manuscript on the investigation hERG1 and integrin interaction and their effect on f-actin organization. Specifically, authors investigate hERG1 expression and localization on the plasma membrane, the role of integrins and Gi proteins in this translocation, how hERG1 and integrin forms a complex, and how the dynamics evolve in the cancer cells, and how f-actin organization is affected. Authors interpretation of the data is supported by the figures presented. The mechanism the authors suggest for the signaling pathway for the actin organization controlled by hERG1-integrin interaction provides new insights on understanding the migration of cancer cells.

Authors provide detailed answers and explanations to reviewers' comments. Accordingly, the manuscript text has been updated, new experiments are included, figures are modified as suggested. The previous version of the manuscript was well written in most parts, and this updated version clarifies the previous questions raised by the reviewers.

Considering all, the manuscript is scientifically sound and clear in my opinion, and contributes to advancing the field. I would suggest the publication of this manuscript.

Below are very minor comments.

- Line 193: typo on girdin
- Section "Insights from the mathematical modeling of experimental data": Authors refer to Fig9 instead of Fig10.
- Authors provided some of the details of the mathematical model in the supplement. It's almost entirely the equations they use. A brief description within the supplement might help, such as how they are solving the rate equations, mentioning that they are using stochastic Monte Carlo for parameter fitting etc. Although such descriptions are written in the methods section in the main text, reiterating with a few sentences in the supplement might be useful.

Dear Editor,

Attached please find the revised version of the manuscript entitled "**Integrins regulate hERG1 dynamics by girdin-dependent G α i3: signaling and modeling in cancer cells**".

We have revised the manuscript according to Reviewers' and Editor comments.

Followings are the point by point answers to the comments:

Reviewer #1:

- the labels of the X axes of the bar graphs in Figure 8, panels A and B, have been added.
- we have indicated that the color code (black/green and red) shown in panel B applies to the whole Figure 8.

Reviewer #2:

- we have amended the typo (girding instead of girdin, line 193) and the wrong reference to the figures in the section "Insights from the mathematical modeling of experimental data" (Fig.8 and 9 instead of Fig.9 and 10, respectively).
- we have added a brief description of the mathematical model (e.g., mentioning that we are using the stochastic Monte Carlo for parameter fitting) in the Supplemental Material pdf file entitled "*Whole-cell network model of hERG1 synthesis, cytoplasmic trafficking and complex formation with integrins on the plasma membrane*"

These changes are **highlighted in red and bold**.

Editor's comments:

- Figure 4A: the blot is now the same as that shown in the Source Data
- Figure 4B: the boxes indicating the areas zoomed in have been placed in the right position
- Figure S2A 3rd panel and S6D 1st panel now show the correct images
- a new blot is now in Fig S4A, as suggested by the Editor
- the labeling in Fig S4A now matches the figure
- Fig S6C and Fig S11A now show the right loading controls, matching what is shown in the Source Data
- Fig S7C and Fig S7D now show their respective right images
- a scale bar has been added to Fig S11C

October 20, 2023

RE: Life Science Alliance Manuscript #LSA-2023-02135-TRR

Prof. Annarosa Arcangeli
University of Florence
Department of Experimental and Clinical Medicine, Section of Internal Medicine, Viale Morgagni 50
Firenze 50134
Italy

Dear Dr. Arcangeli,

Thank you for submitting your revised manuscript entitled "Integrins regulate hERG1 dynamics by girdin-dependent G α i3: signaling and modeling in cancer cells". We would be happy to publish your paper in Life Science Alliance pending final revisions necessary to meet our formatting guidelines.

- please add the Twitter handle of your host institute/organization as well as your own or/and one of the authors in our system
- Supplementary figures should be numbered consecutively (1, 2, 3, 4). Please label your Figure S11 as S9 and correct its callout, etc.
- please be sure to add callouts for all supplementary figures and their panels to your manuscript text

A. FINAL FILES:

B. MANUSCRIPT ORGANIZATION AND FORMATTING:

Sincerely,

October 23, 2023

RE: Life Science Alliance Manuscript #LSA-2023-02135-TRRR

Prof. Annarosa Arcangeli
University of Florence
Department of Experimental and Clinical Medicine, Section of Internal Medicine
Viale Morgagni 50
Firenze 50134
Italy

Dear Dr. Arcangeli,

Thank you for submitting your Research Article entitled "Integrins regulate hERG1 dynamics by girdin-dependent G α i3: signaling and modeling in cancer cells". It is a pleasure to let you know that your manuscript is now accepted for publication in Life Science Alliance. Congratulations on this interesting work.

DISTRIBUTION OF MATERIALS:

Again, congratulations on a very nice paper. I hope you found the review process to be constructive and are pleased with how the manuscript was handled editorially. We look forward to future exciting submissions from your lab.

Sincerely,
